# Reducing sarcolipin expression mitigates Duchenne muscular dystrophy and associated cardiomyopathy in mice

Antanina Voit[1], Vishwendra Patel[2], Ronald Pachon[1], Vikas Shah[1], Mohammad Bakhutma[1], Erik Kohlbrenner[3], Joseph J. McArdle[2], Louis J. Dell'Italia[4], Jerry R. Mendell[5], Lai-Hua Xie[1], Roger J. Hajjar[3], Dongsheng Duan[6], Diego Fraidenraich[1] & Gopal J. Babu[1]

Sarcolipin (SLN) is an inhibitor of the sarco/endoplasmic reticulum (SR) $Ca^{2+}$ ATPase (SERCA) and is abnormally elevated in the muscle of Duchenne muscular dystrophy (DMD) patients and animal models. Here we show that reducing SLN levels ameliorates dystrophic pathology in the severe dystrophin/utrophin double mutant ($mdx:utr^{-/-}$) mouse model of DMD. Germline inactivation of one allele of the SLN gene normalizes SLN expression, restores SERCA function, mitigates skeletal muscle and cardiac pathology, improves muscle regeneration, and extends the lifespan. To translate our findings into a therapeutic strategy, we knock down SLN expression in 1-month old $mdx:utr^{-/-}$ mice via adeno-associated virus (AAV) 9-mediated RNA interference. The AAV treatment markedly reduces SLN expression, attenuates muscle pathology and improves diaphragm, skeletal muscle and cardiac function. Taken together, our findings suggest that SLN reduction is a promising therapeutic approach for DMD.

[1] Department of Cell Biology and Molecular Medicine, New Jersey Medical School, Rutgers, The State University of New Jersey, Newark, NJ 07103, USA. [2] Department of Pharmacology, Physiology and Neuroscience, New Jersey Medical School, Rutgers, The State University of New Jersey, Newark, NJ 07103, USA. [3] Cardiovascular Research Center, Icahn School of Medicine at Mount Sinai, New York, NY 10029, USA. [4] Department of Medicine, University of Alabama at Birmingham, and Birmingham VA Medical Center, Birmingham, AL 35294, USA. [5] Department of Pediatrics and Department of Neurology, Ohio State University Research Institute at Nationwide Children's Hospital, Columbus, OH 43205, USA. [6] Department of Molecular Microbiology and Immunology, Neurology, Bioengineering, Biomedical Sciences, The University of Missouri, Columbia, MO 65212, USA. Correspondence and requests for materials should be addressed to G.J.B. (email: babugo@njms.rutgers.edu)

Duchenne muscular dystrophy (DMD), the most common childhood, severe form of muscular dystrophy[1, 2] is an X-linked disease caused by deficiency of dystrophin protein in muscle[3–5]. DMD patients experience gradual muscle weakness in early childhood and are non-ambulant by 12 years of age. The progressive nature of this disease leads to restrictive pulmonary function, diaphragm dysfunction and cardiomyopathy for which there is no effective treatment[6–8]. The current therapeutic strategies aimed to either replace or compensate for the lack of dystrophin also face major challenges such as targeting cardiac and respiratory tissues and fibrosis[9–12]. Therefore, identification of new therapeutic targets based on disease mechanism could complement the existing strategies and enhance the effectiveness of treatment for this lethal disease.

Accumulating evidence suggests that abnormal elevation of intracellular $Ca^{2+}$ ($Ca^{2+}_i$) is an important, early pathogenic event that initiates and perpetuates disease progression in DMD[13, 14]. Among the several mechanisms that cause $Ca^{2+}_i$ overload[13, 15–20], decreased sarco/endoplasmic reticulum (SR) $Ca^{2+}$ ATPase (SERCA) activity has been considered as a primary cause of $Ca^{2+}_i$ overload in DMD[21–23], because SERCA accounts for ≥70% of $Ca^{2+}$ removal from the cytosol[24, 25]. Accordingly, overexpression of SERCA pump[26–28] or stabilizing the SERCA activity[29] has been shown to ameliorate muscle pathology in mouse models of DMD. However, the molecular basis for SERCA dysfunction in dystrophic muscles remains unknown.

We recently found that sarcolipin (SLN), an inhibitor of SERCA, is abnormally high in the diaphragm and slow- and fast-twitch skeletal muscles of dystrophin mutant (mdx) and dystrophin/utrophin double mutant (mdx:utr−/−) mouse models of DMD[30]. Nevertheless, it is not clear whether SLN upregulation underlies SERCA dysfunction and contributes to the muscle pathogenesis in DMD. In the present study, we sought to determine the physiological relevance of SLN upregulation in DMD following loss-of-function approach by germline gene deletion and by adeno-associated virus (AAV) mediated gene silencing in mdx:utr−/− mice. Our results show that reduction in SLN expression is sufficient to improve the SERCA function and ameliorate the features of muscular dystrophy and cardiomyopathy in mdx:utr−/− mice. Moreover, reducing SLN expression extends the lifespan of mdx:utr−/− mice. These findings provide the first direct evidence on the critical role of SLN upregulation in DMD pathogenesis and identified SLN as a new potential therapeutic target for the treatment of DMD.

## Results

**SLN is upregulated in cardiac and skeletal muscles of DMD.** We recently reported that SLN is abnormally high in the diaphragm and slow- and fast-twitch skeletal muscles of mouse models of DMD[30]. To determine whether SLN upregulation is a common molecular change in skeletal muscles and in the heart in DMD, we analyzed SLN protein levels in the ventricles of DMD mice. Results show that SLN levels were abnormally high in the ventricles of both mdx and mdx:utr−/− mice. Furthermore, SLN levels were ~2 fold higher in the ventricles of mdx:utr−/− mice compared with that of mdx mice (Supplementary Fig. 1a, b). The expression levels of SERCA2a and phospholamban (PLN) were unaltered in the dystrophic hearts. However, the rate of $Ca^{2+}$ dependent $Ca^{2+}$ uptake (Supplementary Fig. 1c) and the maximum velocity ($V_{max}$) of $Ca^{2+}$ uptake (Supplementary Fig. 1d) were significantly reduced; whereas the $EC_{50}$ of $Ca^{2+}$ uptake was unaltered in the ventricles of DMD mice (Supplementary Fig. 1e). Similar changes were observed in the skeletal muscle of the canine DMD model. SLN protein expression was significantly increased in the extensor carpi ulnaris (ECU) muscles of affected dogs

compared to that of non-DMD controls (Supplementary Fig. 2a, b). Further, SERCA 1 levels were significantly increased and SERCA2a levels were decreased in these muscles (Supplementary Fig. 2a, b). Regardless of the changes in SERCA isoforms, the rate of $Ca^{2+}$ dependent $Ca^{2+}$ uptake (Supplementary Fig. 2c) and the $V_{max}$ of $Ca^{2+}$ uptake (Supplementary Fig. 2d) were significantly reduced in dystrophic dog muscles. The $EC_{50}$ for $Ca^{2+}$ activation was not significantly different between the non-DMD and DMD dog tissues (Supplementary Fig. 2e) indicating that the $Ca^{2+}$ affinity of the SERCA pump was not altered. Similar to animal models, SLN levels were increased both in the quadriceps (Supplementary Fig. 3a, b) and in the ventricles (Supplementary Fig. 3c, d) of DMD patients. These findings revealed SLN upregulation as a common molecular change in dystrophin-deficient skeletal and cardiac muscles of both DMD patients and DMD models.

**Ablation of SLN extends the lifespan of mdx:utr−/− mice.** To determine the role of SLN upregulation in DMD, we generated SLN haploinsufficient mdx:utr−/− knockout (mdx:utr−/−:sln +/−) and SLN deficient mdx:utr−/− (mdx:utr−/−:sln−/−) triple knockout (tKO) mice. The mdx:utr−/−:sln+/− and tKO pups were delivered normally. The body weight of mdx:utr−/−:sln+/− and tKO mice were normalized (Fig. 1a) and their lifespan was significantly extended (Fig. 1b; $p < 0.0001$ by the nonparametric log-rank test). The median survival was increased by 446 and 358 days, respectively for mdx:utr−/−:sln+/− and tKO mice compared to that of mdx:utr−/− littermates (73 days). These findings suggest that SLN reduction or ablation markedly improves survival to mdx:utr−/− mice.

**Reducing SLN expression restores SERCA function in DMD.** Deletion of one allele of the SLN gene improved the rate (Fig. 1c) and $V_{max}$ (Fig. 1d) of SR $Ca^{2+}$ uptake in the diaphragm of mdx:utr−/−mice close to that of WT mice. Complete loss of SLN resulted in a further increase in the rate of $Ca^{2+}$ uptake in the tKO diaphragm; however, the $V_{max}$ of $Ca^{2+}$ uptake was not statistically different from the WT controls (Fig. 1d). Furthermore, the $EC_{50}$ for $Ca^{2+}$ activation was significantly decreased in the diaphragm of both mdx:utr−/−:sln+/− and tKO mice (Fig. 1e), indicating an increase in the apparent affinity of SERCA pump for $Ca^{2+}$ ions in SLN deficient dystrophic muscles. Together, these findings suggest that SLN upregulation is the major cause of SERCA dysfunction in dystrophic muscles.

The diaphragm and fast-twitch muscles from mdx:utr−/− mice showed differential expression of SERCA isoforms and increased levels of calsequestrin (CSQ)[30]. We therefore determined whether SLN ablation affected the expression levels of these proteins in dystrophic diaphragm, quadriceps and pectoral muscles. Deletion of one SLN allele reduced the SLN protein expression close to that of WT controls in diaphragm, pectoral and quadriceps muscles (Fig. 1f–h). Further, our results showed that reduction or ablation of SLN restored the expression of SERCA isoforms, as well as normalized CSQ levels in these muscles (Fig. 1f–h). Taken together, these findings suggest that SLN haploinsufficiency is sufficient to normalize SLN expression and reinstate SR function in dystrophic muscles.

**Reducing SLN expression ameliorates muscle pathophysiology.** We next determined whether the improvement in $Ca^{2+}_i$ cycling via enhanced SERCA function reduced the $Ca^{2+}$ dependent protease activity and prevented muscle damage. Reduction or ablation of SLN expression attenuated the calpain activity in pectoral muscle, a representative and severely affected dystrophic muscle (Fig. 2a). Histopathological analysis (Fig. 2b) and

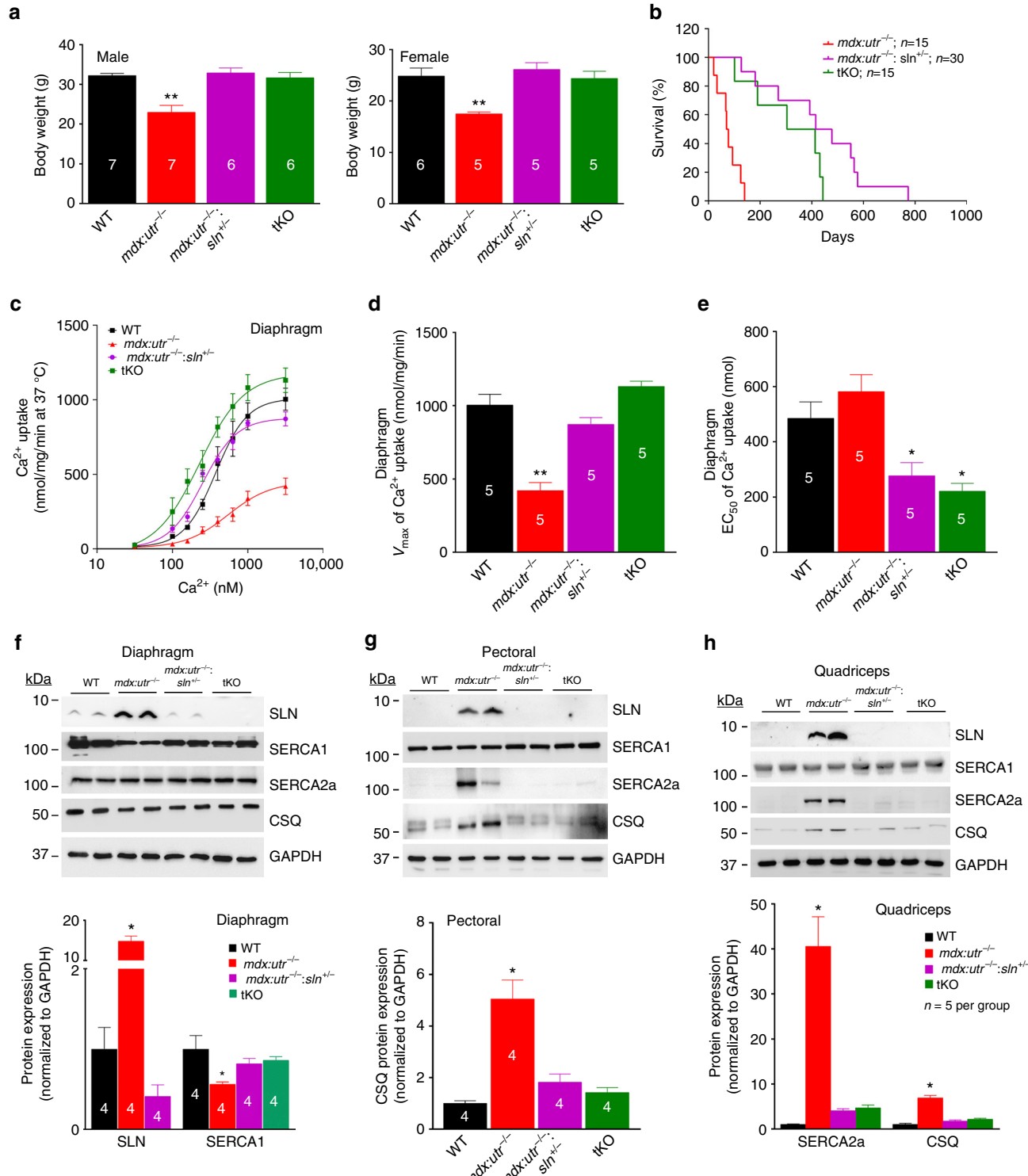

**Fig. 1** Reduction in SLN expression prolongs the lifespan of $mdx:utr^{-/-}$ mice. **a** The body weight of $mdx:utr^{-/-}:sln^{+/-}$ and tKO mice in comparison to that of age- and sex- matched WT and $mdx:utr^{-/-}$ control mice. We used 3–4 month old mice for this experiment. The n number for each group is shown within the bar. Data are presented as mean ± SEM. **significantly different from other groups ($p < 0.005$, t-test with Welch's correction). **b** Kaplan–Meier survival curves indicate that the $mdx:utr^{-/-}:sln^{+/-}$ and tKO mice has increased lifespan in comparison to that of $mdx:utr^{-/-}$ controls as determined by the nonparametric log-rank test; $p < 0.0001$. **c** Increased $Ca^{2+}$ dependent SR $Ca^{2+}$ uptake, **d** increased $V_{max}$ of $Ca^{2+}$ uptake and **e** decreased $EC_{50}$ values in the diaphragm of $mdx:utr^{-/-}:sln^{+/-}$ and tKO mice indicates improved SERCA function. The n number for each group is shown within the bar. Data are presented as mean ± SEM. **significantly different from other groups ($p < 0.0005$, t-test with Welch's correction). *significantly different from WT and $mdx:utr^{-/-}$ mice ($p < 0.005$, t-test with Welch's correction). **f–h** Representative western blot analysis and quantitation of SLN, SERCA1, SERCA2a and CSQ proteins in the diaphragm, pectoral and quadriceps muscles respectively. Tissues are from 3 to 4 month old mice. Uncropped scans of western blots are shown in Supplementary Fig. 8a–c. Data are represented as mean ± SEM. The n number for each group is shown within the bar diagram. *significantly different from other groups, $p < 0.05$, t-test with Welch's correction

quantitation show significant reduction in necrosis and fibrosis in both diaphragm (Fig. 2c, d) and quadriceps (Fig. 2e, f) of *mdx: utr*$^{-/-}$*:sln*$^{+/-}$ mice and in the quadriceps of tKO mice. These improvements were less prominent in the tKO diaphragm and were not statistically different from the *mdx:utr*$^{-/-}$ controls. We next determined the muscle fiber size in quadriceps by measuring the minimal Feret's diameter variance coefficient (VC) following wheat germ agglutinin (WGA) staining (Fig. 3a). The VC was significantly increased in the muscles of *mdx:utr*$^{-/-}$ mice indicating heterogeneity in fiber size. On the other hand, in the muscles of *mdx:utr*$^{-/-}$*:sln*$^{+/-}$ and tKO mice, the VC was significantly reduced indicating reduction in small regenerating split fibers and hypertrophic fibers. These findings prompted us to determine whether SLN ablation has any effect on the muscle regeneration process as well as fiber-type transition. Immunostaining and quantitation showed that fibers expressing embryonic myosin heavy chain (eMyHC) or Type I MyHC were significantly higher in the pectoral muscles of *mdx:utr*$^{-/-}$ mice (Fig. 3b) and was consistent with our previous findings on the dystrophic quadriceps[30]. In contrast, the number of fibers expressing these proteins were significantly decreased in the muscles of *mdx: utr*$^{-/-}$*:sln*$^{+/-}$ and tKO mice (Fig. 3b). These findings suggest that reduction in SLN expression can improve the muscle regeneration process as well as prevent the fiber-type transition in dystrophic muscles.

We next investigated the effect of SLN ablation on muscle mechanics. The forelimb muscle grip strength was significantly increased in the *mdx:utr*$^{-/-}$*:sln*$^{+/-}$ and tKO mice compared to that of *mdx:utr*$^{-/-}$ littermates (Fig. 4a). We extended these studies by measuring the isometric contractile properties of the dystrophic extensor digitorum longus (EDL; a less severely affected) and diaphragm (a more severely affected) muscles. The twitch-tension (Fig. 4b, c), 10–90% rising slope (rate of contraction) and 90–10% decay slope (rate of relaxation) at 2 Hz (Supplementary Fig. 4a, b) and force-frequency curves (Fig. 4d) were significantly increased in the EDL muscle of *mdx: utr*$^{-/-}$*:sln*$^{+/-}$ mice. These contractile parameters were also increased in the EDL of tKO mice but not at a statistically significant level. The half-maximal force stimulation frequency for EDL remained unaltered among the experimental groups (WT = $27 \pm 3$ (n = 7), *mdx:utr*$^{-/-}$ = $28 \pm 3$ (n = 5), *mdx:utr*$^{-/-}$: *sln*$^{+/-}$ = $26 \pm 2$ (n = 8) and tKO = $28 \pm 1$ (n = 5) Hz; unpaired *t*-test with Welch's correction). The effect of SLN ablation on diaphragm function in the *mdx:utr*$^{-/-}$ mice was less prominent. The twitch-tension at 2 Hz was significantly increased in the hemidiaphragm of *mdx:utr*$^{-/-}$*:sln*$^{+/-}$ but not in the tKO mice (Fig. 4e, f). The 10–90% rising slope and 90–10% decay slope obtained from the hemidiaphragm of *mdx:utr*$^{-/-}$*:sln*$^{+/-}$ and tKO at 2 Hz showed a slightly increasing trend but were not significantly different from that of *mdx:utr*$^{-/-}$ controls (Supplementary Fig. 4c, d). Similarly, the force-frequency curves for the hemidiaphragm from *mdx:utr*$^{-/-}$*:sln*$^{+/-}$ was shifted upwards but to a smaller extent in the tKO mice (Fig. 4g). The half-maximal force stimulation frequency for hemidiaphragm remained unaltered among all four mice groups (WT = $11 \pm 0.3$ (n = 6), *mdx:utr*$^{-/-}$ = $13 \pm 0.8$ (n = 5), *mdx:utr*$^{-/-}$*:sln*$^{+/-}$ = $12 \pm 0.5$ (n = 6) and tKO = $14 \pm 1.2$ (n = 6) Hz; unpaired *t*-test with Welch's correction). These functional data corroborated the differences and structural improvements seen in the diaphragm of *mdx:utr*$^{-/-}$*:sln*$^{+/-}$ and tKO mice (Fig. 2b–f). These findings suggest that reduction in SLN expression is sufficient to improve the functional properties of dystrophic skeletal muscles.

**Ablation of SLN ameliorates dystrophic cardiomyopathy.** We next determined whether SLN ablation ameliorates cardiac

pathology in the *mdx:utr*$^{-/-}$ mice. Western blot analysis (Supplementary Fig. 5a) and quantitation showed that loss of one SLN allele was sufficient to reduce SLN protein expression in atria (*mdx:utr*$^{-/-}$ = $1.7 \pm 0.2$ vs. *mdx:utr*$^{-/-}$*:sln*$^{+/-}$ = $0.9 \pm 0.1$ fold; n = 4; p < 0.05, *t*-test with Welch's correction) and in the ventricles of *mdx:utr*$^{-/-}$ mice near to WT levels. SLN ablation had no effect on the expression levels of SERCA2a, PLN, ryanodine receptor (RyR), and CSQ (Supplementary Fig. 5a). H&E and trichrome staining and quantitation showed that mononuclear infiltration and fibrosis were significantly reduced in the *mdx:utr*$^{-/-}$*:sln*$^{+/-}$ and tKO ventricles compared to that of *mdx:utr*$^{-/-}$ controls (Fig. 5a). Cardiac function evaluated by echocardiography (Supplementary Fig. 5b) showed a marked improvement in left ventricular (LV) function as evident from the increased LV ejection fraction (EF; Fig. 5b) and fractional shortening (FS; Fig. 5c) in the *mdx:utr*$^{-/-}$*:sln*$^{+/-}$ and tKO mice. There was an increase in interventricular septal end systole (IVSs) and posterior wall thickness along with significant reduction in LV internal diameter end diastole (LVIDd) in the *mdx:utr*$^{-/-}$: *sln*$^{+/-}$ and tKO mice (cTable 1). These findings suggested that hearts from these mice undergo specific concentric remodeling that contributes to the improved cardiac function. These findings indicate that normalizing SLN level is sufficient to preserve cardiac function and mitigate dystrophic cardiomyopathy in mice.

**Postnatal AAV9 SLN sh*RNA* gene therapy mitigates DMD.** Findings from the above studies suggest that normalizing SLN expression is sufficient to mitigate the severe DMD phenotype including muscle pathophysiology, diaphragm function and cardiomyopathy. To translate these findings into a therapeutic strategy, we knocked down SLN expression postnatally in 1-month old *mdx:utr*$^{-/-}$ mice via AAV9 mediated expression of SLN specific short-hairpin RNA (sh*SLN*). The AAV9.sh*SLN* treatment for 12 weeks significantly reduced SLN expression in both skeletal muscle (0.22-fold vs. saline treated controls; p < 0.05) and LV (similar to WT) of *mdx:utr*$^{-/-}$ mice (Fig. 6a, b and Supplementary Fig. 6a, b). In the *mdx:utr*$^{-/-}$ myocardium, AAV treatment had no effect on the protein expression of SERCA2a, PLN, CSQ and RyR (Fig. 6b). On the other hand, AAV treatment reduced SERCA2a (p < 0.05) and CSQ (p < 0.07) protein levels in the pectoral muscles of *mdx:utr*$^{-/-}$ mice (Fig. 6a and Supplementary Fig. 6a). These data are consistent with the findings on the *mdx:utr*$^{-/-}$*:sln*$^{+/-}$ mice (Fig. 1g) and suggest that postnatal reduction in SLN expression can also restore the SR function in the *mdx:utr*$^{-/-}$ mice.

We next determined, whether AAV gene therapy mitigates cardiac and skeletal muscle pathophysiology in the *mdx:utr*$^{-/-}$ mice. The outcome of these studies mimics the data from the *mdx:utr*$^{-/-}$*:sln*$^{+/-}$ mice. H&E staining (Supplementary Fig. 6c) and quantitation showed that the invasion of mononuclear cells and cell necrosis were significantly reduced both in the skeletal muscle (Fig. 6c) and in the ventricles (Fig. 6d) of AAV treated *mdx:utr*$^{-/-}$ mice. AAV treatment also improved the LV systolic function and cardiac remodeling (Fig. 6e and Table 2) in *mdx:utr*$^{-/-}$ mice. Forelimb muscle strength was significantly improved in AAV treated *mdx:utr*$^{-/-}$ mice (Fig. 6f). Furthermore, the twitch-tension (Fig. 6g–h), 10–90% rising slope and 90–10% decay slope at 2 Hz and force–frequency relationships, (Supplementary Fig. 7a–f) were significantly increased in both EDL and hemidiaphragm of AAV treated groups indicating improved muscle mechanics. Altogether these findings suggest that AAV mediated postnatal reduction in SLN expression is sufficient to mitigate the severe muscular dystrophy and associated cardiomyopathy in *mdx:utr*$^{-/-}$ mice.

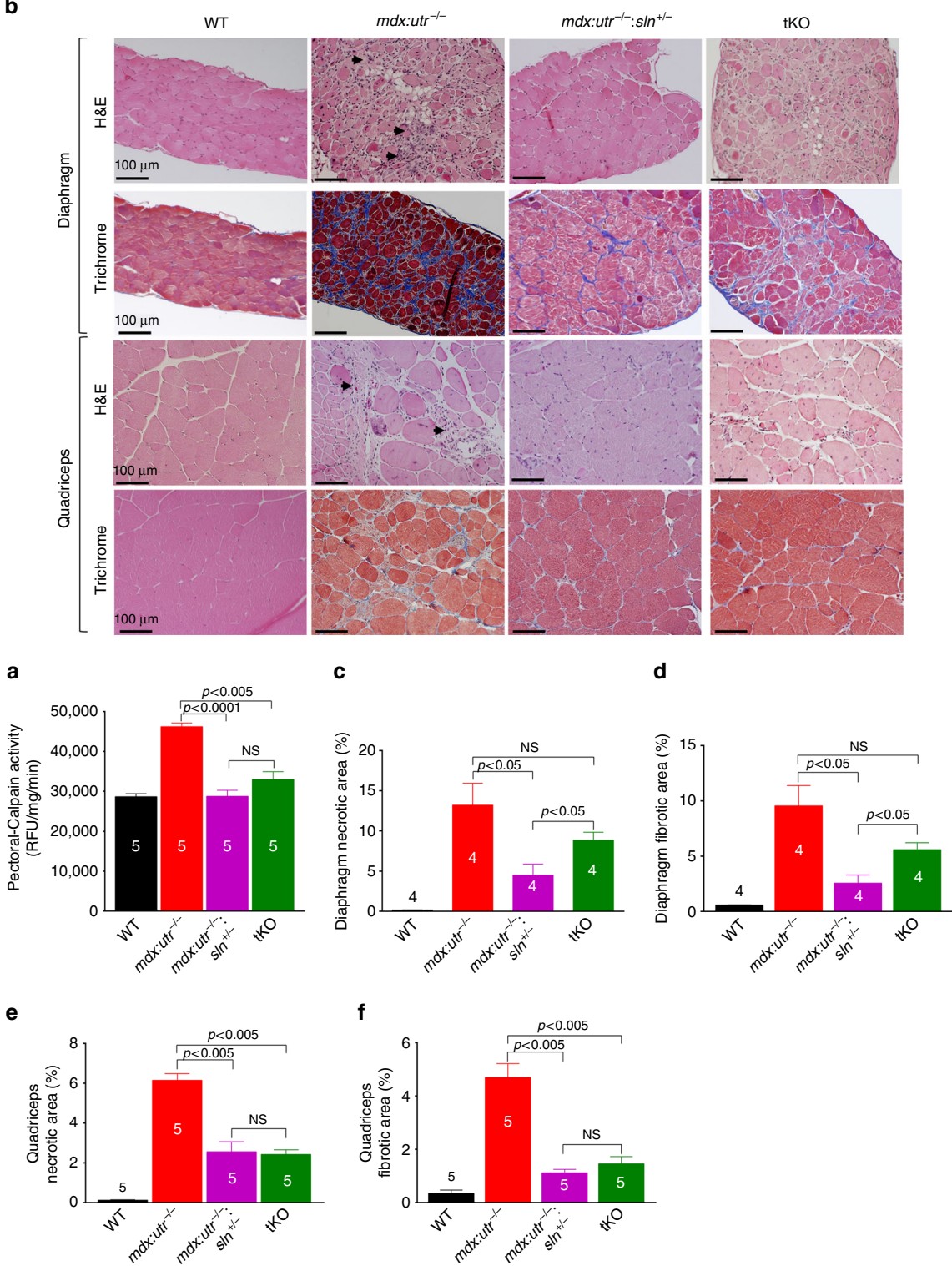

**Fig. 2** Reduction in SLN expression ameliorates muscle pathology. **a** Calpain activity is restored to normal levels in the pectoral muscles of *mdx:utr⁻/⁻:sln⁺/⁻* and tKO mice. Data are presented as mean ± SEM (*t*-test with Welch's correction) of five independent experiments performed in duplicates. The *n* number for each group is shown within the bar. **b** Representative H&E and Masson's trichrome stained quadriceps and diaphragm muscles. Arrow indicates increased mononuclear infiltration (indicative of necrosis) and collagen (blue) accumulation (indicative of fibrosis) in *mdx:utr⁻/⁻* mice. Original magnification is ×20. Scale bar=100 μm. **c–f** Quantitation show that the necrotic and fibrotic areas were significantly reduced in both diaphragm and quadriceps of *mdx:utr⁻/⁻:sln⁺/⁻* mice and in the quadriceps of tKO mice in comparison to that of *mdx:utr⁻/⁻* controls. The *n* number for each group and the *p* values (*t*-test with Welch's correction) are shown within the graph. Data are presented as mean ± SEM. Tissues from 3 to 4 month old mice are used for all the above experiments

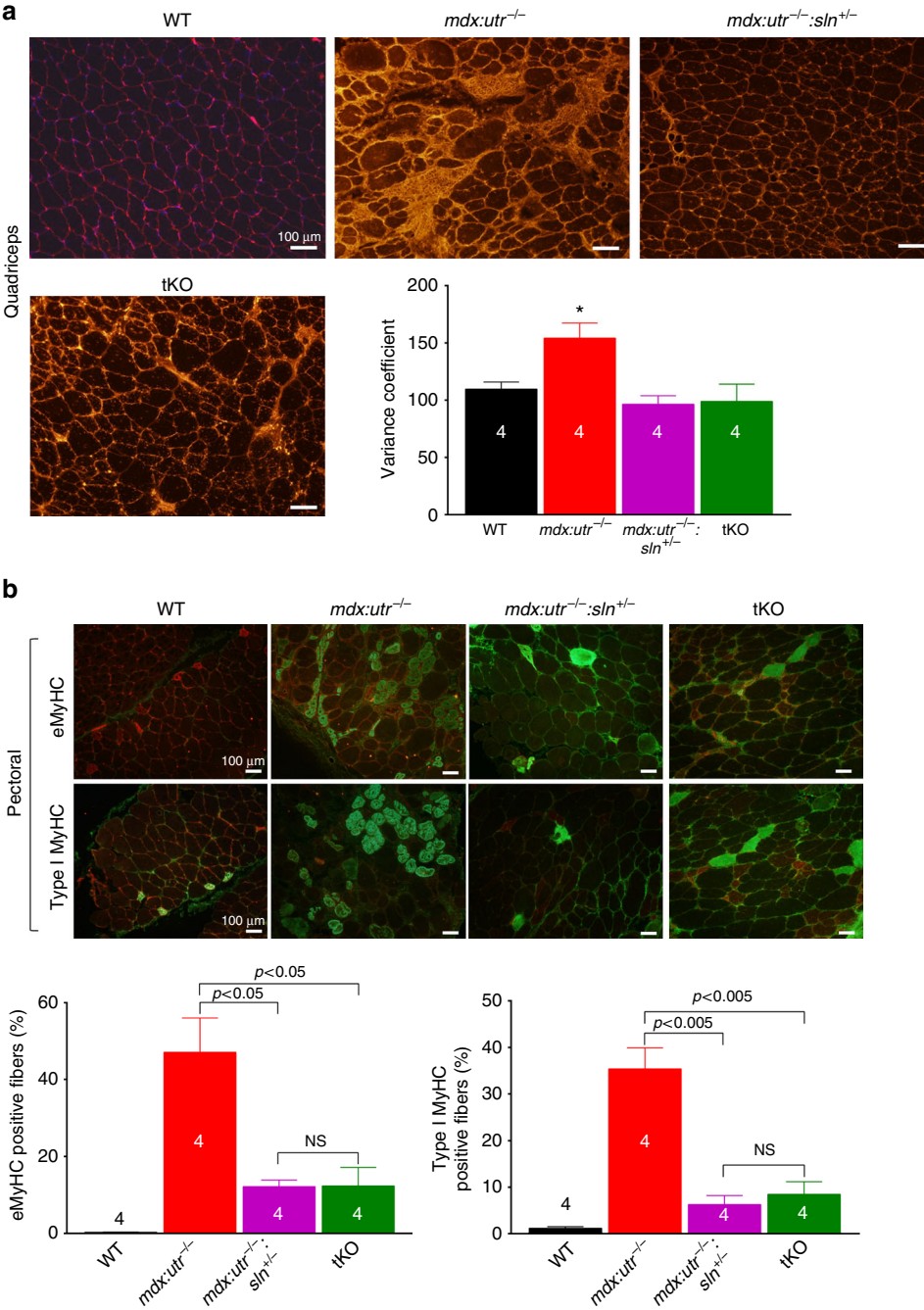

**Fig. 3** Reduction in SLN expression improves muscle regeneration and prevents fiber-type transition. **a** Representative images of WGA stained sections of quadriceps used for fiber size measurements. Original magnification is ×10. The measurements of minimal "Feret's" diameter variance coefficients (VC) of the muscle fiber size shown in the bar graph indicates that VC of fiber size is significantly reduced in the quadriceps of *mdx:utr*$^{-/-}$:*sln*$^{+/-}$ and tKO mice. Data are presented as mean ± SEM of four independent experiments. *significantly different from other groups ($p < 0.05$, $t$-test with Welch's correction). Scale bar=100 μm. **b** Representative images of pectoral muscle sections immunostained for eMyHC (green) or Type I MyHC (green) and stained with WGA (red). Original magnification is ×10. Scale bar=100 μm. The quantitation of muscle fibers positive for eMyHC and Type I MyHC are shown in the bar graph. The *n* number for each group and the *p* values ($t$-test with Welch's correction) are shown within the graph. Data are presented as mean ± SEM. NS-not statistically significant. Tissues from 3 to 4 month old mice are used for all the above experiments

## Discussion

We have previously shown that SLN protein levels are abnormally elevated in the diaphragm and slow- and fast-twitch skeletal muscles of mouse models of DMD[30]. Here, we further show that SLN upregulation is a common molecular change in both skeletal muscle and heart in murine and canine DMD models and human patients. Furthermore, the SERCA function as measured by SR Ca$^{2+}$ uptake is also significantly decreased in both skeletal and cardiac muscles of DMD models. These findings led to the hypothesis that increased SLN protein expression could chronically inhibit SERCA pump and cause sustained elevation of Ca$^{2+}$$_i$ levels, which subsequently contribute for the activation of Ca$^{2+}$ dependent proteases and tissue remodeling, and muscle pathogenesis in DMD. Accordingly, reducing SLN protein levels

is anticipated to improve SERCA function and mitigate DMD. This hypothesis was tested following a loss of function approach in $mdx:utr^{-/-}$, a severe and lethal mouse model of DMD. As predicted, our findings suggest that reduction in SLN expression is sufficient to improve the SERCA function in dystrophic muscles. Furthermore, reduction in SLN protein expression ameliorated the severe muscular dystrophy phenotype, and extended the lifespan of $mdx:utr^{-/-}$ mice.

SERCA function in muscle is regulated by three small molecular weight SR membrane proteins: PLN, SLN, and myoregulin

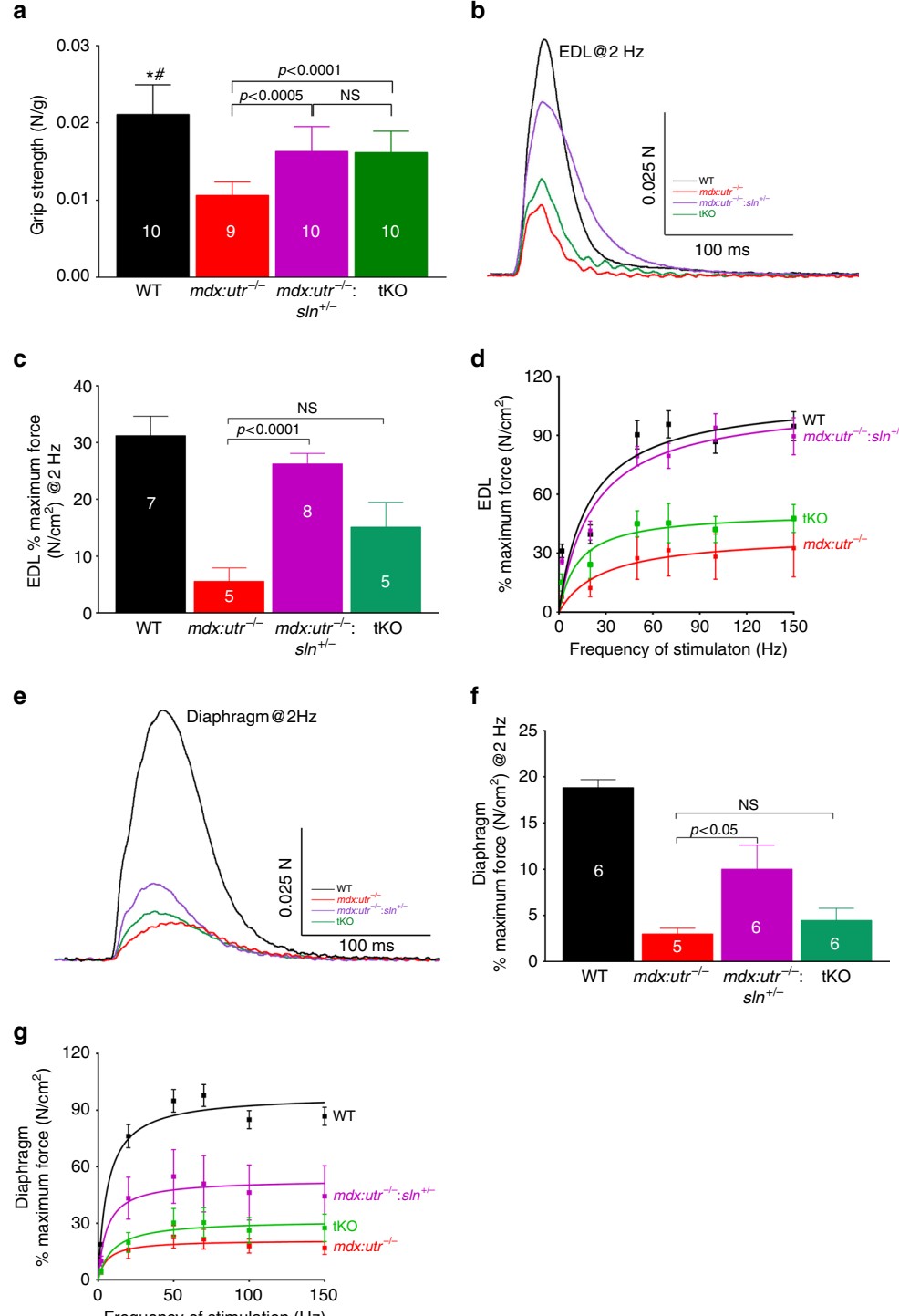

**Fig. 4** Reduction in SLN expression improves skeletal muscle function in $mdx:utr^{-/-}$ mice. **a** Forelimb strength measured using a grip strength meter shows improved muscle strength in the $mdx:utr^{-/-}:sln^{+/-}$ and tKO mice. We used 3–4 month old male and female mice for this study. The n number for each group is shown within the bar. Data are presented as mean ± SEM (t-test with Welch's correction). # $p < 0.0001$ vs. $mdx:utr^{-/-}$. *$p < 0.05$ vs. $mdx:utr^{-/-}:sln^{+/-}$ and tKO mice. **b, e** Representative traces of twitch force at 2 Hz for EDL and hemidiaphragm respectively. **c, f** The maximum force generated by the EDL and hemidiaphragm at 2 Hz are significantly increased in the $mdx:utr^{-/-}:sln^{+/-}$ mice compared to that of $mdx:utr^{-/-}$ mice. The n number for each group is shown within the bar. Data are presented as mean ± SEM. **d, g** Force-frequency curves indicating that force generated by EDL and hemidiaphragm in the $mdx:utr^{-/-}:sln^{+/-}$ mice are significantly increased at all frequencies. EDL and hemidiaphragm are from 3 to 4 month old mice

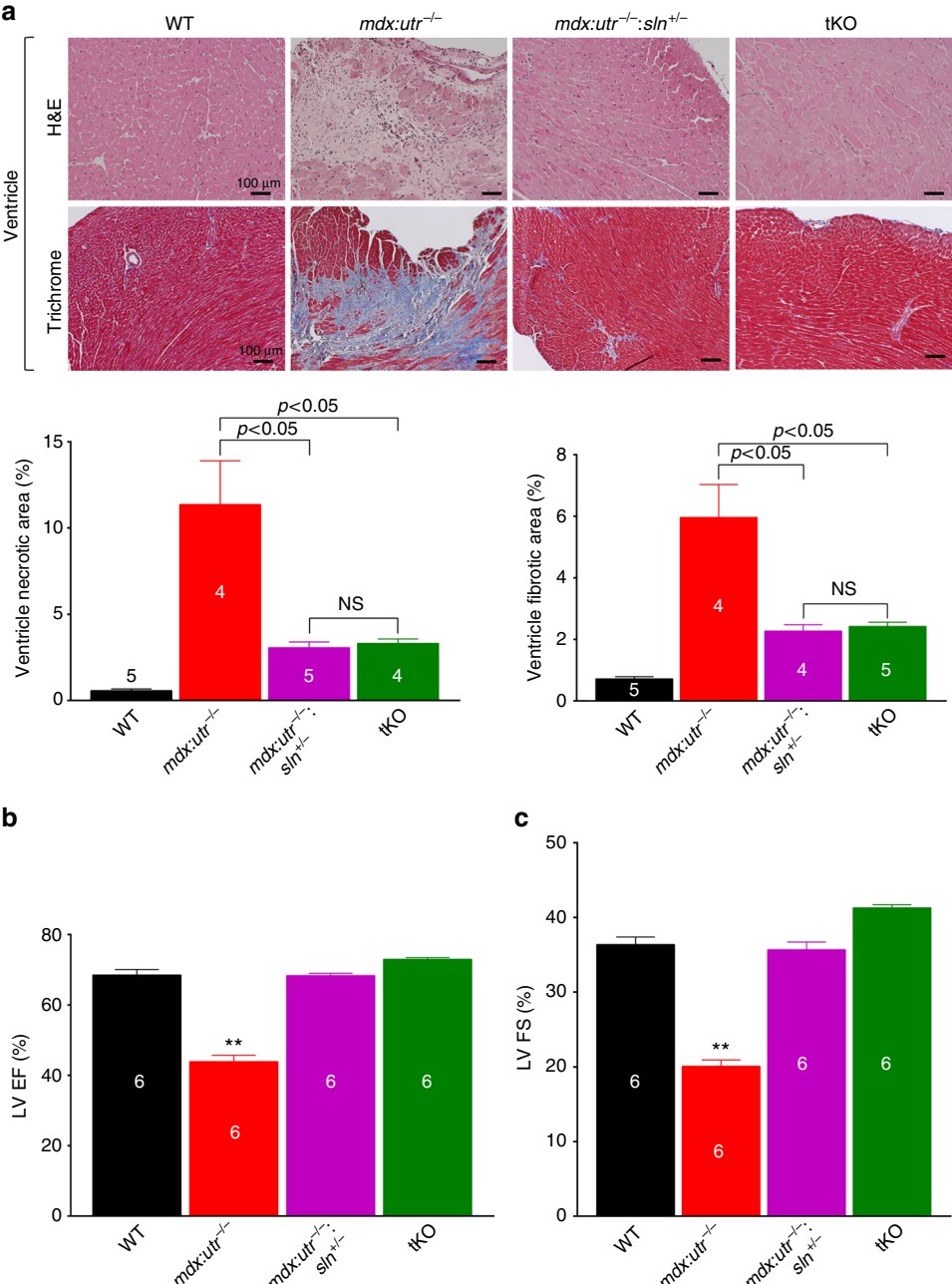

**Fig. 5** Ablation of SLN expression prevents cardiomyopathy in *mdx:utr*$^{-/-}$ mice. All cardiac studies are performed in 3–4 month old mice. **a** Representative H&E and Masson's trichrome stained ventricular tissue sections. Original magnification is ×10. Scale bar=100 μm. Bar graph indicate that reduction or ablation of SLN significantly reduced fibrosis and necrosis in the *mdx:utr*$^{-/-}$ hearts. The *n* number for each group and the *p* values (*t*-test with Welch's correction) are shown within the graph. Data are presented as mean ± SEM. **b**, **c** Echocardiographic measurements demonstrate that the LV ejection fraction (EF) and fractional shortening (FS) are restored to normal values in the *mdx:utr*$^{-/-}$:*sln*$^{+/-}$ and tKO mice. **p* < 0.0001 vs. other groups; *n* = 6 per group. Data are presented as mean ± SEM (*t*-test with Welch's correction)

(MLN). In rodents, MLN is primarily expressed in skeletal muscles[31, 32]; whereas SLN is predominantly expressed in the tongue followed by diaphragm and slow-skeletal muscles but not expressed in the fast-twitch skeletal muscles[32, 33]. In contrast, in larger mammals SLN is expressed in all skeletal muscle tissues that have been evaluated[33]. In the heart, PLN expression is high in the ventricles[32, 33], while SLN expression is very high in atria and very low in the ventricles of both rodents and larger mammals[33]. Findings from the present study along with our earlier report demonstrate that in DMD, SLN but not PLN is elevated in the diaphragm, and slow- (soleus) and fast-twitch (quadriceps and pectoral muscle) skeletal muscles, as well in atria and in the ventricles. Our findings are also consistent with the microarray and semi-quantitative RT-PCR data, which show SLN mRNA upregulation in the medial gastrocnemius of *mdx* mice[34]. Altogether these findings suggest that SLN upregulation is a common molecular change in all skeletal muscle tissues and in the heart in DMD.

In this study, we did not investigate whether MLN expression is altered and it has a role in SERCA function in DMD. We were

**Table 1 Baseline echocardiographic data of 3–4 month old mice**

|  | WT | mdx:utr$^{-/-}$ | mdx:utr$^{-/-}$:sln$^{+/-}$ | tKO |
|---|---|---|---|---|
| IVSd (mm) | 0.74 ± 0.05 | 0.83 ± 0.07 | 0.96 ± 0.05$^{\$}$ | 0.85 ± 0.07 |
| IVSs (mm) | 1.28 ± 0.06 | 0.99 ± 0.05* | 1.39 ± 0.09 | 1.345 ± 0.08 |
| LVIDd (mm) | 4.2 ± 0.13 | 3.5 ± 0.24$^{\$}$ | 3.34 ± 0.13$^{\$}$ | 3.25 ± 0.06$^{\$}$ |
| LVIDs (mm) | 2.68 ± 0.07 | 2.81 ± 0.20 | 2.215 ± 0.10$^{++}$ | 1.91 ± 0.03$^{++}$ |
| LVPWd (mm) | 0.63 ± 0.05 | 0.76 ± 0.07 | 0.78 ± 0.03 | 0.83 ± 0.09 |
| LVPWs (mm) | 1.13 ± 0.03 | 0.83 ± 0.06* | 1.07 ± 0.05 | 1.21 ± 0.09 |
| FS (%) | 36 ± 1.0 | 20 ± 0.8** | 36 ± 1.0 | 41 ± 0.4$^{\#}$ |
| EF (%) | 69 ± 1.5 | 44 ± 1.8** | 68 ± 0.6 | 73 ± 0.4$^{\#}$ |
| HR (bpm) | 466 ± 16 | 498 ± 52 | 531 ± 47 | 529 ± 25 |

IVSd interventricular septal end diastole, IVSs interventricular septal end systole, LVIDd left ventricular internal diameter end diastole, LVIDs left ventricular internal diameter end systole, LVPWd left ventricular posterior wall end diastole, LVPWs left ventricular posterior wall end systole, FS fractional shortening, EF ejection fraction, HR heart rate
$^{\$}p < 0.005$ vs. WT; $^{*}p < 0.05$ vs. other groups; $^{**}p < 0.0001$ vs. other groups; $^{\#}p < 0.05$ vs. WT & mdx:utr$^{-/-}$:sln$^{+/-}$; $^{++}p < 0.05$ vs. WT & mdx:utr$^{-/-}$; $n = 6$ per group. Data are presented as mean ± SEM

also unable to demonstrate the absolute stoichiometry of SLN/SERCA, and its relation to SR $Ca^{2+}$ uptake in various dystrophic muscles. These studies are necessary to validate the direct role of SLN in SERCA function in various dystrophic muscles. Nevertheless, the current study suggests that SLN upregulation could be a major cause of SERCA dysfunction and elevation of $Ca^{2+}_i$ in DMD. The increased $V_{max}$ of $Ca^{2+}$ uptake and the increased apparent binding affinity of SERCA pump for $Ca^{2+}$ (as shown by decreased $EC_{50}$) in the mdx:utr$^{-/-}$:sln$^{+/-}$ and tKO muscles supports this view. The possible indirect effect of improved $Ca^{2+}_i$ cycling, including reduced $Ca^{2+}$ dependent protease activity and decreased fibrosis and necrosis in muscles of mdx:utr$^{-/-}$:sln$^{+/-}$ and tKO mice further supports our hypothesis.

Studies from Periasamy's laboratory have demonstrated that uncoupling of SERCA by SLN is an important regulator of adaptive thermogenesis and diet induced diabetes by regulation of mitochondrial function[35, 36]. In the context of the DMD, they also demonstrated higher oxygen consumption relative to activity or skeletal muscle force generation in mdx:utr$^{-/-}$ mice[37], which could be due to either an inefficiency in mitochondrial energy utilization or a decrease in energy production. This is consistent with the markedly abnormal mitochondria by transmission electron microscope demonstrating multiple small mitochondria with decreased electron density and more sparsely spaced cristae in a highly disorganized mitochondrial sarcomeric structure in mdx:utr$^{-/-}$ mice compared to normal architecture[37]. Increased $Ca^{2+}_i$ level has been linked to increased reactive oxygen species (ROS) production and mitochondrial damage, as well as the inflammatory pathways in DMD[38]. These changes can further lead to vicious cycle of increased mitochondrial ROS production, bioenergetic dysfunction, altered nuclear signaling and post-translational modification of proteins, including activation of proteases such as calpain. We therefore speculate that reduction in SLN expression could prevent mitochondrial-derived oxidative stress and inflammatory pathways by reducing the $Ca^{2+}_i$ load and contribute to the mitigation of severe muscular dystrophy phenotype in mdx:utr$^{-/-}$ mice.

Altered expression of SLN at the mRNA and protein levels have been reported in the diseased myocardium of human[39–42] and in animal models[33, 43, 44], as well as in mouse models with skeletal muscle pathology[45–48]. However, the functional consequences of altered SLN expression in cardiac and skeletal muscle pathology are not fully understood. A recent study shows that complete loss of SLN fails to improve SERCA function and a centronuclear myopathy-like phenotype in transgenic mice with skeletal muscle specific PLN overexpression[45]. On the other hand, SLN ablation results in exacerbated muscle atrophy and weakness with impaired calcineurin pathway, suggesting that SLN overexpression may have compensatory effects on muscle function in

these mice[45]. We did not find any possible explanation for this discrepancy with current findings in DMD mice. However, our data also suggest that total loss of SLN is not beneficial to dystrophin deficient skeletal muscles. It is also important to note that SLN deficient mice (sln$^{-/-}$) are normal as evident by no change in body weight, survival and quality of life[35, 49, 50]. These mice also show improved atrial[49] and skeletal muscle[50] function. Here, we show that either reduction or total loss of SLN equally improved the LV function and reduced the myocardial fibrosis and necrosis in mdx:utr$^{-/-}$ mice. These findings suggest that the beneficial effect of total SLN loss is more pronounced in the heart than skeletal muscles in the mdx:utr$^{-/-}$ mice. Therefore, it is possible that the improved cardiac function can partly contribute to the improvement in quality of life and survival of both mdx:utr$^{-/-}$:sln$^{+/-}$ and tKO mice. Similar to dystrophic cardiomyopathy, SLN is also elevated in the ventricles of patients with mitral regurgitation[40] and in patients with Tako-Tsubo cardiomyopathy[41]. SLN could be a novel therapeutic target, if the observed beneficial effects of SLN downregulation in DMD-cardiomyopathy can be recapitulated in these diseases.

The detailed regulatory mechanisms which control SLN protein levels in various dystrophic muscles are not fully understood. Our data show that loss of one SLN allele can restore SLN protein levels close to that of WT levels instead of a graded 50% reduction in all dystrophic muscles analyzed. This could be due to changes in transcription efficiency, transcript stability, translation efficiency, post-translational modification and protein half-life in the tissues of mdx:utr$^{-/-}$:sln$^{+/-}$ mice. Re-expression of SERCA2a isoform and increased CSQ in dystrophic fast-twitch muscles such as quadriceps and pectoral muscle may be a compensatory alteration to prevent deficiencies in SERCA function. Our findings suggest that improving SERCA function upon SLN reduction is sufficient to revert these changes and reinstate the SR function in dystrophic muscles. In addition, our findings suggest that reduction in SLN expression can improve the muscle regeneration process and prevent fiber-type transition in dystrophic muscles, although the mechanistic basis for this improvement remains unclear. Maintenance of SR $Ca^{2+}$ content, an important determinant of $Ca^{2+}_i$ level[51–53] is vital to differentiation and myoblast fusion[54]. Similarly, it has been reported that increased $Ca^{2+}_i$ level induces slow-fiber transformation of fast-twitch muscles[55, 56]. Therefore, one possible explanation is that germline reduction or ablation of SLN expression in satellite cells could normalize the $Ca^{2+}_i$ cycling and improve myoblast fusion and differentiation as well as prevent fiber-type transition. In support of our hypothesis, stable C2C12 cell line expressing SLN delays the activation of genes involved in muscle differentiation[52].

Germline gene knockouts are often associated with compensatory alterations and these changes may influence the

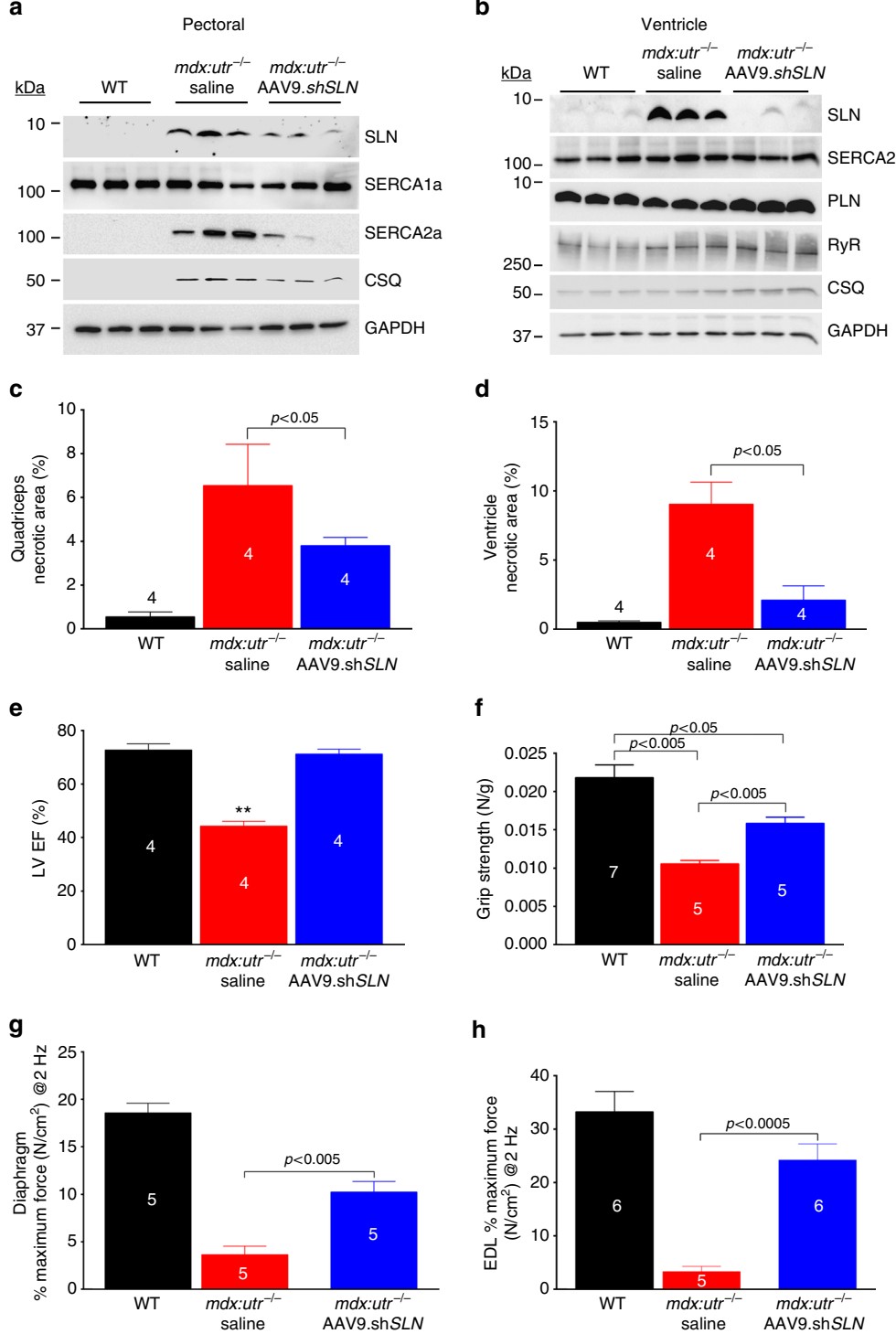

**Fig. 6** Postnatal AAV9.sh*SLN* treatment ameliorates DMD and associated cardiomyopathy in *mdx:utr*$^{-/-}$ mice. 1-month old male and female mice are injected with AAV or saline and experiments are performed 12 weeks post-injection. **a**, **b** Representative western blots show that AAV9.sh*SLN* treatment effectively reduces SLN expression in skeletal (pectoral muscle) and cardiac (ventricle) muscles. Uncropped scans of western blots are shown in Supplementary Fig. 8d, e. The AAV treatment reduced SERCA2a and CSQ levels in the pectoral muscles of *mdx:utr*$^{-/-}$ mice. **c**, **d** Quantitation of areas with mononuclear infiltration in the H&E stained tissue sections show that cell necrosis is significantly reduced in both quadriceps and ventricles of AAV treated *mdx:utr*$^{-/-}$ mice compared to that of saline injected *mdx:utr*$^{-/-}$ controls. Data are presented as mean ± SEM. The *n* number for each group and the *p* values (*t*-test with Welch's correction) are shown within the graph. **e** LV EF, **f** forelimb muscle strength, and **g**, **h** the maximum twitch force generated by EDL and hemidiaphragm at 2 Hz are significantly improved in the AAV treated mice compared to that of saline injected *mdx:utr*$^{-/-}$ controls. \*\**p* < 0.0001 vs. other groups. Data are presented as mean ± SEM. The *n* number for each group and the *p* values (*t*-test with Welch's correction) are shown within the graph

**Table 2 Baseline echocardiographic data of AAV treated *mdx:utr*⁻/⁻ mice**

|  | WT | *mdx:utr*⁻/⁻ Saline | *mdx:utr*⁻/⁻ AAV9.sh*SLN* |
|---|---|---|---|
| IVSd (mm) | $0.90 \pm 0.04$ | $0.95 \pm 0.14$ | $0.78 \pm 0.08$ |
| IVSs (mm) | $1.31 \pm 0.04$ | $0.98 \pm 0.08^*$ | $1.3 \pm 0.08$ |
| LVIDd (mm) | $4.06 \pm 0.11$ | $3.1 \pm 0.24^\$$ | $3.12 \pm 0.1^\$$ |
| LVIDs (mm) | $2.58 \pm 0.07$ | $2.43 \pm 0.20$ | $1.89 \pm 0.09^\#$ |
| LVPWd (mm) | $0.77 \pm 0.08$ | $0.72 \pm 0.06$ | $0.9 \pm 0.09$ |
| LVPWs (mm) | $1.12 \pm 0.06$ | $0.89 \pm 0.05^*$ | $1.02 \pm 0.09$ |
| FS (%) | $37 \pm 0.6$ | $22 \pm 1.0^{**}$ | $40 \pm 1.4$ |
| EF (%) | $73 \pm 2.3$ | $44 \pm 1.8^{**}$ | $71 \pm 1.8$ |
| HR (bpm) | $449 \pm 27$ | $585 \pm 7^\$$ | $559 \pm 58$ |

IVSd interventricular septal end diastole, IVSs interventricular septal end systole, LVIDd left ventricular internal diameter end diastole, LVIDs left ventricular internal diameter end systole, LVPWd left ventricular posterior wall end diastole, LVPWs left ventricular posterior wall end systole, FS fractional shortening, EF ejection fraction, HR heart rate.
$^\$p < 0.05$ vs. WT; $^*p < 0.05$ vs. other groups; $^{**}p < 0.005$ vs. other groups; $^\#p < 0.06$ vs. other groups. $n = 4$ per group. Data are presented as mean ± SEM

**Table 3 Tissues used for various biochemical, histopathological and functional analyses**

| Name of the assay | Tissues used |
|---|---|
| Western blot analysis | Atria, ventricles, diaphragm, quadriceps and pectoral muscle |
| Histopathology | Quadriceps, diaphragm and ventricles |
| Immunostaining | Pectoral muscle |
| Calpain assay | Pectoral muscle |
| Muscle physiology | EDL and diaphragm |

experimental outcome. Therefore, it is important to determine whether reducing SLN expression during postnatal period can have the same beneficial effect. Furthermore, preclinical studies are required for validating the potential of targeting SLN expression/function in future clinical trials. As a first step towards this goal, we reduced SLN expression in postnatal period in *mdx:utr*⁻/⁻ mice using AAV expressing short-hairpin RNA specific for SLN. AAV9 serotype has been successfully adopted in preclinical studies to reduce gene expression in cardiac and skeletal muscles[57–60]. Therefore, we chose AAV9 to express sh*SLN* in *mdx:utr*⁻/⁻ mice. Systemic injection of AAV9.sh*SLN* reduced SLN expression in both heart and skeletal muscles. Furthermore, AAV treatment has similar beneficial effects including restoration of SERCA isoforms in fast-twitch muscles and amelioration of diaphragm and skeletal muscle pathophysiology and cardiomyopathy in DMD. Together these findings suggest that AAV mediated SLN gene silencing might represent a novel gene therapy approach to treat DMD.

In summary, this study provides evidence that SLN upregulation is a molecular basis for SERCA dysfunction in both skeletal and cardiac muscles of DMD. Furthermore, we have provided initial evidence that SLN could be a common and potent therapeutic target for the treatment of diaphragm and skeletal muscle pathology and cardiomyopathy in DMD. Our studies also raise the possibility that reducing SLN expression or function by AAV or other strategies can provide therapeutic benefits for DMD.

## Methods

**Animal studies.** All experimental procedures involving mice in this study were approved by the Institutional Animal Care and Use Committee (IACUC) of New Jersey Medical School, Rutgers, Newark, NJ. The tissue samples from normal and DMD dogs were from studies approved by the IACUC of University of Missouri, Columbia, MO to Dr. Dongsheng Duan.

**Mice.** We used 3–4 months old male and female mice in C57BL/6 background for all the experiments described in this study. The *mdx:utr*⁺/⁻ mice, a gift from Dr. Robert Grange (Virginia Tech) (to Dr. Diego Fraidenraich), from a line originally derived and obtained from Drs. Mark Grady and Joshua Sanes (Washington University)[61]. The *mdx:utr*⁻/⁻:*sln*⁺/⁻ and tKO (*mdx:utr*⁻/⁻:*sln*⁻/⁻) mice were generated by crossing the *mdx:utr*⁺/⁻ mice to *sln*⁻/⁻ mice[49]. These mice were crossed for five generations to obtain the *mdx:utr*⁺/⁻:*sln*⁺/⁻ mice in isogenic background. The male and female *mdx:utr*⁺/⁻:*sln*⁺/⁻ mice were then crossed to generate the *mdx:utr*⁻/⁻, *mdx:utr*⁻/⁻:*sln*⁺/⁻ and tKO mice. Mice were kept under a 12-h light/dark cycle with a temperature of 22–24 °C and 60–70% of humidity and fed ad libitum with normal chow diet. The genotypes of the mice were identified by PCR analysis using previously published sequences[49, 62]. Animal numbers were predetermined based on pilot studies and sample sizes were similar to generally employed in the field. No samples, mice or data points were excluded from the data analysis. Animals were not randomized except for the genotypes. For

echocardiography and muscle force measurements, investigators were blinded for the genotypes.

The tissues used for various biochemical, histopathological and functional analyses are shown in Table 3.

**Experimental dogs.** All dog-related experiments were approved by the IACUC of the University of Missouri and were performed in accordance with NIH guidelines. All the experimental dogs were on a mixed genetic background consisting of golden retriever, labrador retriever, Corgi and beagle and generated in house at the University of Missouri by artificial insemination. Affected dogs carry various mutations in the dystrophin gene that abort dystrophin expression. The genotype was determined by polymerase chain reaction according to published protocols[63, 64]. The diagnosis was confirmed by the significantly elevated serum creatine kinase level in affected dogs. All experimental dogs were housed in a specific-pathogen free animal care facility and kept under a 12-h light/12-h dark cycle. Affected dogs were housed in a raised platform kennel while normal dogs were housed in regular floor kennel. Depending on the age and size, two or more dogs are housed together to promote socialization. Normal dogs were fed dry Purina Lab Diet 5006, while affected dogs were fed wet Purina Proplan Puppy food. Dogs were given ad libitum access to clean drinking water. Toys were allowed in the kennel with dogs for enrichment. Dogs were monitored daily by the caregiver for overall health condition and activity. A full physical examination was performed by the veterinarian from the Office of Animal Research at the University of Missouri for any unusual changes (such as behavior, activity, food and water consumption, and clinical symptoms). We used both male and dogs for this study. The non-DMD controls were in the age range of 1.73 to 31 month old and the DMD dogs were in the age range of 6.8 to 11.9 month old. Experimental subjects were euthanized at the end of the study according to the 2013 AVMA Guidelines for the Euthanasia of Animals. Freshly dissected ECU muscle was snap frozen in liquid nitrogen in blocks of ~0.5–1 cm³. Frozen muscle tissues were kept in −80 °C freezer until use.

**Human samples.** Two non-DMD and two DMD male human ventricular samples[65] obtained from the University of Maryland Brain and Tissue Bank, a member of the NIH NeuroBioBank network were used for this study. All samples were dissected post-mortem. DMD1 cause of death was attributable to cardiac failure at age 15; while DMD2 cause of death was attributed to pulmonary thromboembolism at age 17. The research use of these samples was approved by the Institutional Review Board (IRB) at Rutgers New Jersey Medical School (to Dr. Diego Fraidenraich).

The research use of the human quadriceps tissues was approved by the IRB at the Ohio State University/Nationwide Children's Hospital (to Dr. Jerry R. Mendell, MD) and performed in accordance with relevant guidelines and regulations. The two non-DMD human quadriceps were from a 4-year-old to 6-year-old normal child. The two DMD quadriceps were from 11 to 15-year-old DMD patients. Informed consent was obtained from all subjects from whom tissues were analyzed.

**AAV9.sh*SLN* generation and delivery into *mdx:utr*⁻/⁻ mice.** The sh*SLN* sequence (ACTTCACAGTTGTCCTCATCACTCGAGTGATGAGGACAACTGT GAAG) was first cloned into plasmid pds-sh*PLB*[57] using HindIII and BamHI sites to replace the sh*PLB* sequence. The entire 975 bp AAV cassette (ITR-U6-promoter-shSLN-bgHpA-ITR) and also some plasmid backbone regions flanking the ITRs were digested out with BspHI and inserted into pFastBacDual at the NcoI site. The resulting plasmid "pFB-ITR-sh-SLN" was used to create a recombinant Baculovirus "Bac-ITR-sh-SLN" using the Bac-to-Bac System (Invitrogen). The AAV9.sh*SLN* was produced in Sf9 insect cells as previously described[66]. Briefly, Sf9 insect cell lines were infected with Bac-ITR-sh-SLN, BacRep and BacCap9 (kindly provided by Sergei Zolotukhin) viruses. After 3 days post-infection, cells were collected and the AAV was purified by iodixanol gradient ultracentrifugation and dialyzed into Lactated Ringer's. The viral titers were determined by qPCR using primers binding the bGH region (forward: 5′TGCCTTCCTTGACCCT; reverse: 5′ CCTTGCTGT CCTGCCC) and dilutions of the AAV2 Reference Standard Material (ATCC) to generate a standard curve.

For AAV9.sh*SLN* injection studies, we used 1-month old male and female *mdx:utr*$^{-/-}$ mice. Mice were divided into two groups: the AAV-treated group and the saline treated group. We used 6 mice per group. The AAV9.sh*SLN* vector ($1 \times 10^{11}$ genome) was delivered to the mice via a single bolus tail vein injection. The mice were sacrificed 16 weeks of age after measuring the forelimb strength and cardiac function by M-mode echocardiography and the tissues were used for functional and biochemical studies.

**Isometric force measurements**. Isometric force in isolated muscle tissues was determined as described[67]. Briefly, hemidiaphragm and EDL were harvested immediately following euthanasia and kept in cold oxygenated Ringer's solution (in mmol/l, 135 NaCl, 5 KCl, 1 MgCl$_2$, 2 CaCl$_2$, 1 Na$_2$HPO$_4$, 15 NaHCO$_3$, and 5.5 glucose). Hemidiaphragm and EDL preparations were mounted in a Rodnoti chamber (Rodnoti Glass Technology, Inc., CA, USA) containing oxygenated (95% O$_2$–5% CO$_2$) Ringer's solution at room temperature (~ 22 °C). One tendon was attached to the bottom of the Rodnoti chamber while the other tendon was attached to a Grass force transducer. For direct stimulation of muscles, two parallel plate (1 × 1 cm$^2$) silver electrodes that were attached to the inner wall of Rodnoti chamber to deliver 1 ms long square electric pulses produced by a Grass stimulator. Muscles were adjusted to the optimal length (Lo) for force generation. At the end of the protocol, muscle length and weight were measured using a Vernier caliper and weighing scale respectively. To detect the threshold stimulation, the stimulation voltage was increased till maximal force of contraction was achieved. Trains of 40 suprathreshold (120% of threshold) stimuli ranging from 2 to 150 Hz were applied to study the muscle force generation. The response signals were digitized (Digidata 1440A Axon Instruments), acquired, and analyzed using PCLAMP software (version 10.1, Axon Instruments). Muscle cross sectional area (CSA) for force normalization was calculated using the following formula as described before[68]. CSA = $m/(Lo \times L/Lo \times 1.06$ mg/mm$^3$), where "$m$" is muscle mass (in mg), Lo is optimal muscle length, $L/Lo$ is ratio of fiber length to muscle length (0.45 for EDL and 1 for diaphragm) and 1.06 mg/mm$^3$ is muscle density.

**Histological analysis**. Five micron paraffin sections of various skeletal and cardiac tissues from WT, *mdx:utr*$^{-/-}$, *mdx:utr*$^{-/-}$:*sln*$^{+/-}$ and tKO mice were stained with Hematoxylin and Eosin (H&E) and Masson's trichrome following standard procedures. The red stained collagen areas by trichrome staining indicating fibrosis and necrotic areas containing mononuclear cells stained by H&E were calculated using NIH ImageJ 1.43u program.

**Immunofluorescence**. The mouse monoclonal antibodies specific for eMyHC (BF45) and type 1 MyHC (BAF8) were purchased from Developmental Studies Hybridoma Bank. Tissues were cryo-sectioned at 10 μm and immunostained using antibodies specific for eMyHC (1:10) or type 1 MyHC (1:5) overnight at 4 °C and processed as before[69]. For fiber size measurements, the tissue sections were stained with WGA, (fluorophore conjugated, 1:100, Cat.# W11262, Life Technologies). Images were obtained using a Zeiss LSM 510 on Zeiss Axiovert 100 M Base and processed using NIS Elements. The minimal "Feret's" diameter variance coefficients of the muscle fiber size was calculated on the WGA stained sections using the ImageJ 1.43u program.

**Western blot analysis**. Mouse tissues were dissected out, rinsed in sterile PBS, and flash frozen in liquid nitrogen. Mouse, dog or human tissue homogenization was performed in lysis buffer (in mmol/l, 50 Tris, pH 7.4, 150 NaCl, 1 EDTA, and 0.5% NP-40) supplemented with PMSF (1 mmole), NaVO$_3$(5 mmole), Okadaic Acid (10 nmole), NaF (1 mmole), and benzamidine (1 mmole). Equal amounts of total protein extracts were separated on the sodium dodecyl sulfate-polyacrylamide gels (SDS–PAGE) along with pre-stained molecular weight markers and transferred to nitrocellulose membranes for 1 h at room temperature. After transfer, the membranes were stained with Ponceau S and cut into small stripes based on the molecular weight of each protein studied. The membrane strips were then blocked with 3% milk in phosphate-buffered saline, and probed overnight at 4 °C using antibodies specific for SLN (anti-rabbit, 1:3000)[33], SERCA1 (anti-rabbit, 1:2000, custom made)[33], SERCA2a (anti-rabbit, 1:5000, custom made)[33], PLN (anti-rabbit, 1:3000, custom made)[33], CSQ (anti-rabbit, 1:5000, Affinity Bio Reagents) which recognizes both cardiac and skeletal isoforms, RyR (anti-mouse, 1:1000, Affinity BioReagents) or GAPDH (anti-mouse, 1:10,000, Sigma). Membranes were incubated with appropriate secondary antibodies for 45 min at room temperature and visualized with SuperSignal West Dura Substrate kit (ThermoFisher Scientific) using Bio-Rad ChemiDoc MP Imaging system. Quantitation of signals were performed using Image Lab version 5.1 software and then normalized to GAPDH levels. Western blots were repeated at least three times. Uncropped scans of the western images are shown in Supplementary Figs. 8 and 9.

**Grip-strength measurements**. An assessment of muscle function was recorded using grip strength meter (Columbus Instruments). The grip strength meter was positioned horizontally and the mouse was held by its tail and allowed to securely grip the triangular pull bar. After the mouse obtained a solid grasp of the triangular pull bar, the mouse was pulled backward parallel to the device. The force that was applied to the bar at the time of release was recorded as the peak grip strength

(Newton). This was repeated three times and an average force was determined for each mouse. Grip strength values were normalized by the weight (g) of each animal to get the grip strength (N/g) ratio.

**SR Ca$^{2+}$ uptake**. SR Ca$^{2+}$ uptake was measured following the Millipore filtration technique, as previously described[69]. Briefly, about 150 μg of the total protein extract was incubated at 37 °C in 1.5 ml of Ca$^{2+}$ uptake medium (in mmol/l, 40 imidazole, pH 7.0, 100 KCl, 5 MgCl$_2$, 5NaN$_3$, 5 potassium oxalate, and 0.5 EGTA) and various concentrations of CaCl$_2$ to yield 0.03–3 μmol/l free Ca$^{2+}$ (containing 1 μCi/μmol$^{45}$Ca$^{2+}$). To obtain the maximal stimulation of SR Ca$^{2+}$ uptake, ruthenium red was added to a final concentration of 1 μmole immediately prior to the addition of the substrates to begin the Ca$^{2+}$ uptake. The reaction was initiated by the addition of ATP to a final concentration of 5 mmole and terminated at 1 min by filtration. Each assay was performed in duplicate. The rate of SR Ca$^{2+}$ uptake and the Ca$^{2+}$ concentration required for EC$_{50}$ were determined by non-linear curve fitting analysis using GraphPad Prism v6.01 software.

**Calpain assay**. Activated calpain in the protein extract was measured using a calpain activity assay kit (Abcam, Cat.# ab65308). Briefly, cytosolic protein extracts of pectoral muscle were prepared with the extraction buffer which prevents the auto-activation of calpain during the extraction procedure. The calpain activity was measured using fluorometric assay using calpain substrate Ac-LLY-AFC. The activity was represented as relative fluorescence units (RFU)/mg protein.

**Echocardiography**. Mice were anesthetized with 2.5% tribromoethanol and echocardiography was performed using the high resolution ultrasound machine VisualSonic/Vevo 770 system with a high frequency transducer (30 MHz) as described[69]. Left ventricular (LV) dimensions, wall thicknesses, LV fractional shortening (FS), and LV ejection fraction (EF) were measured from the LV M-Mode images.

**Statistical analysis**. We followed the established DMD-standard operating procedures for outcome measurements for fiber-size, quantitation of fibrosis and necrosis and muscle mechanics (http://www.treat-nmd.eu/research/preclinical/dmd-sops/). All statistical analyses were performed using GraphPad Prism v6.01 software. Results are presented as the mean ± SEM. Differences were determined using a two-tailed, unpaired Student's "$t$" test with Welch's correction. Two-way analysis of variance (ANOVA) with post-hoc Bonferroni correction were used for multi-group comparison when necessary. A value of $p < 0.05$ was considered as significant. The survival curve was generated using Kaplan–Meier survival analysis and data was analyzed using log-rank (Mantel–Cox) test.

**Data availability**. The data reported in this study are available from the corresponding author upon reasonable request.

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

## Acknowledgements

This work was supported by the National Institute of Arthritis and Musculoskeletal and Skin Diseases, US National Institutes of Health (NIH) grant (Grant #1R01AR069107 to G.J.B.), Rutgers, New Jersey Medical School Dean's Biomedical Research bridge-grant (to G.J.B.) and partially supported by the American Heart Association (Founders Affiliates) grant in aid (Grant # 16GRNT30960034 to G.J.B.), NIH grant (Grant # 1R01AR070517 to D.D. and G.J.B.) and Jesse's Journey foundation grant (to D.D. and G.J.B.). The content is solely the responsibility of the authors and does not necessarily represent the official views of the National Institutes of Health.

## Author contributions

G.J.B. conceived the study. A.V. and G.J.B. designed experiments and wrote the manuscript. A.V., V.P., R.P., V.S., M.B., E.K., and G.J.B. performed experiments; A.V., V.P., R.P., J.J.M., L.J.D., J.R.M., L.-H.X., R.J.H., D.D., D.F., and G.J.B. analyzed and critically discussed the data. All authors discussed the results and implications and commented on the manuscript at all stages.

## Additional information

**Competing interests:** The authors declare no competing financial interests.

