## [Peer Review file · Nature Communications]

Reviewers' comments:

Reviewer #1 (Remarks to the Author):

Elevated cytosolic calcium in mdx muscle has previously been shown to be due to increased Ca²⁺ influx (Goonasekera et al, Human Mol Genet (2014) 23:3706-3715), increased SR Ca²⁺ leak (Bellinger et al, Nat. Med. 15:325-330, 2009) and decreased SERCA activity (Divet A and Huchet-Cadiou C. 2002, Pflugers Arch 444: 634-643.). In addition previous workers have shown that increasing SERCA activity improves muscle function in mdx mice (Goonasekera, S.A., et al. J Clin Invest 121, 1044-1052 (2011)). From this standpoint the observation that decreasing the levels of an inhibitor of SERCA (SLN) improves muscle function in these mouse models of DMD is not particularly novel. Likewise the observation that SLN is increased in skeletal muscle of mdx/utr mice was previously reported by the authors of the current paper (Schneider et al, J Muscle Res Cell Motil 34: 349-356. doi: 10.1007/s10974-013-9350-0.). The new findings reported in the current manuscript are: 1) SLN is also increased in cardiac tissue of mdx/utr^{-/-} mice and 2) the dystrophic symptoms in both skeletal and cardiac muscle can be alleviated by decreasing SLN. These data are quite convincing. Hence reducing SLN might be an effective therapeutic strategy for DMD.

There are a number of issues that need to be addressed by the authors.

- 1) This paper does not extend our mechanistic understanding of DMD. For example, why does dystrophin deficiency increase SLN? How does reducing SLN and increasing SERCA activity in the muscle fibers decrease fibrosis?
- 2) The lifespan extension of the heterozygous SLN deficient mdx/utr^{-/-} mice is very impressive. What does a deficiency in SLN do to WT mice? For example, how does SLN deficiency affect lifespan in the presence of dystrophin (Fig. 1b)? What are the consequences to both the mdx/utr^{-/-} mice and dystrophin normal mice with exercise? These issues should at least be discussed.
- 3) What is the significance of the increased CSQ levels in mdx/utr^{-/-} mice?
- 4) The mdx/utr^{-/-}/sln^{-/-} show improved life span but the histology does not appear to be greatly improved in supplemental Fig.3. Does this mean that most of the mdx/utr^{-/-} mice normally die of heart failure? The beneficial effects of SLN deficiency seem to be greater in the heart than in skeletal muscle. This should be discussed. Based on Supplemental Figure 3, it is difficult to see how the data in fig 2b and c with respect to necrosis and fibrosis were obtained. Perhaps the authors need to provide a more representative figure.
- 5) Likewise the panel for eMyHC for the mdx/utr^{-/-} mice does not appear to be very representative of the summarized data (looks a lot more than 40%).
- 6) In figure 3 is the wt grip strength significantly different for the mdx/utr? An n of 7 is very low for grip strength measurements. Are the error bars SEM or SD?
- 7) The force frequency curves in Fig. 3d and e do not look like normal curves for these muscles. The force should be normalized to cross-sectional area rather than muscle weight. Second, the stimulation frequency that produces half maximal force appears to be somewhat low for both EDL and diaphragm from 3-4 month old mice. The data should be fit, maximal force (normalized to CSA), and the half maximal stimulation frequency should be provided. Also, the WT and mdx/utr^{-/-} data in Fig3a and Supplemental figure 6a appear to be the same. Shouldn't the data in Sfig 6 for AAV-shSLN mice be compared to mice receiving an empty AAV9 vector and also use littermate mice?
- 8) The authors need to show representative echocardiograms for each type of mouse rather than just summarized data.
- 9) Fig. 3 b and c and Supplementary Fig 6 should show representative twitch force tracings illustrating the measurements (rising slope and decay slope) summarized in the bar graphs.
- 10) Supplementary fig 2 (the human data) has only an n of 2 seems premature and should be eliminated.
- 11) The authors need to state in the figure legends if the data are mean +/- SEM or SD and state the statistical test used for that particular data set.

Reviewer #2 (Remarks to the Author):

In this report by Voit et al, the roles of sarcolipin on the development of striated muscle structure and function were evaluated in Duchenne muscular dystrophy models.

This is an interesting finding concerning about a target for future therapeutical interventions .

However, several key evidence is missing to support the working model. Therefore, the manuscript, in this format, is not suitable for publication in Nature Communications.

1. Statistical power

The study on animal models needs to be refine. Although largely consistent, the in vivo data are missing 'n'. The authors looked at cardiac function on 'n'=4. This is largely insufficient for this type of study. Given the non-invasive methods performed in this study (grip strength, echocardiography, survival), one could wonder why not all mice have been assessed for all the studied parameters. This is crucial.

2. SLN expression should be assessed along the progression of the disease in the animal models. It is important to understand the pathogenic role of sarcolipin in a pre- and post-sympatomatic stage.

Other points that need to be further addressed:

1. the manuscript needs a deep english correction
2. FigS1, why the authors show 'n'=2 on immunoblots and 'n'=3 on quantification. I found it rather odd. They could have showed all the samples on their SDS-PAGE.
3. Each figure should be on one and only page.
4. the histology staining should be homogenised. The authors should show a lower magnification of the tissue with an inset of their magnification (especially for heart sections)
5. Material Method section is poorly documented. There are no data on dog GRMD samples, the age of the animals are NEVER documented,...

Reviewer #3 (Remarks to the Author):

Summary

Manuscript NCOMMS-17-03326-T by Voit et al. (Reducing Sarcolipin Expression Mitigates Duchenne Muscular Dystrophy and Associated Cardiomyopathy in Mice) is a tour-de-force of molecular physiology and translational research. The authors comprise a strong collaboration of expert investigators in myopathies and cardiovascular disease, including calcium signaling, gene therapy, and cell therapy. The manuscript uses two well-characterized mouse models of DMD (dystrophin-KO = mdx and dystrophin/utrophin-double KO = DKO) for transgenic and AAV approaches to reduce SLN expression (DKO:SLN+/-, triple KO = TKO, and AAV9-shSLN). The manuscript also examines SLN expression in a dog model of DMD and in human DMD patients. The manuscript reports exciting results and identifies SLN as a new therapeutic target for muscular dystrophy and heart failure. The project is well designed and performed. The manuscript would be strengthened by additional detail (If consistent with the length limitations of the journal), and small mistakes need to be fixed.

Six main assays were utilized:

1. SLN and SERCA proteins were detected by Western blot of 6 tissue homogenates: mouse diaphragm, mouse ventricle, mouse atria, mouse pectorals, dog extensor carpi ulnaris (ECU; upper foreleg), and human ventricle.

2. Histology (H&E = necrotic, trichrome = fibrotic, WGA = size) was measured in 4 dissected tissues: mouse quadriceps, mouse diaphragm, mouse pectorals, and mouse ventricle,
3. SR Ca uptake was measured in 3 tissue homogenates: mouse diaphragm, mouse ventricle, and dog ECU.
4. Isometric force was measured in two isolated muscles: mouse quadriceps and mouse extensor digitorum longus (EDL; upper hindlimb), and mouse grip strength was measured.
5. Calpain protease activity was measured in 1 muscle homogenate: mouse pectorals.
6. Cardiac echocardiography in anesthetized mice.

Results indicate that SLN reduction enhances DKO mouse survival, SR Ca uptake, muscle force production, and cardiac contractility, while decreasing muscle calpain activity. Altogether, very promising results.

Reviewer Assessment, per Nat Comm criteria

1. Novelty: High
2. Impact: High
3. Writing: Needs improvement (see Comments 1 below)
3. Human subjects: Incomplete documentation (comment 2)
4. Animal subjects: Incomplete documentation (comment 3)
4. Literature: Incomplete (comments 4, 5)
5. Statistical Analysis: Incomplete (comments 6, 7)
8. Convincing: Yes

Comments

1. Organization: Here we point out problems in organization, but leave it to the Editors to determine whether these are consistent with the standards and length constraints of the journal. The manuscript is 6 page of text, including 'Introductory Paragraph'. There are no section headers or subsections for Results or Discussion. There is a Methods section (5 pages) added at the end of the manuscript; is Methods intended as Supplementary Information? There are two separate reference sections: Manuscript ('Literature Cited') and Methods ('References'). Pages of the manuscript are not numbered.

The Figure legends are long. In the four Figs of the main manuscript, the average length of caption is 190 ± 32 words, ie, all are longer than the Introductory Paragraph (ie, Abstract, which is 129 words). Much of the text in Fig captions may be more appropriate for inclusion in Results or Methods sections. The captions in Supplement are 105 ± 49 words for 6 SI Figs. The main manuscript is short in length because many results are described in Fig captions.

2. More information on is needed on the human muscle samples (quadriceps and heart), per Nat Comm recommended guidelines ('Biospecimen Reporting for Improved Study Quality' @ Cancer Cytopathol 119:92-101, 2011). There is no information on human samples (n= 2 for non-DMD and DMD) in the Methods sections. Please describe how human biopsy samples were harvested, processed, and stored for Western blots (eg, location of biopsies in heart: ventricle or atrium?). Other information which needs to be added are selection process for donors and clinical characterization of DMD patients and healthy volunteers.

3. More information is needed on the dog DMD samples (ECU), per Nature Comm recommended guidelines ('Animals in Research: Reporting In Vivo Experiments' @ PLoS Biol. 8(6):e1000412, 2010). There is no information on dog samples (n= 2 for normal and DMD) in the Methods sections. Please add a sentence and citation for how the dog DMD model was generated. Please describe how dog quadriceps were harvested, processed, and stored for Western blots and activity assays.

4. The manuscript is too brief with little discussion of published reports from the SLN field. Please address the following topics.

4a. The hallmark of SLN-KO mice is that they are acutely-sensitive to cold temperature and

become obese on a high-fat diet (Periasamy, Nature Med 2012, FASEB J 2013, JBC 2015a,b, JBC 2016). In particular, expression level of SLN in mouse soleus depends on cage temperature (25 or 37C). Please describe the feeding protocol, housing temperature, and running wheel availability for mice strains in the current manuscript. Fig 1A demonstrates that reduction and ablation of SLN increases body weight in the DKO background. Please discuss current results in comparison to SLN-KO results from Periasamy lab.

4b. Myoregulin (MLN) has been proposed to be the main regulator of SERCA in mouse muscle (Olson, Cell 2015, Science 2016). According to that paper, SLN protein expression in mouse had been previously detected at significant level only in soleus and diaphragm. The manuscript should discuss the hypothesis from Olson lab that MLN is the main SERCA regulator in mouse muscle.

4c. A relevant article was published during review which reports that SLN deletion exacerbates muscle disease in a different mouse model (ie, SLN-KO is causative, not compensatory). Please discuss this new article in relation to the current manuscript. "Sarcolipin deletion exacerbates soleus muscle atrophy and weakness in phospholamban overexpressing mice. Fajardo VA, PLoS One. 2017 Mar 9;12(3):e0173708. doi: 10.1371/journal.pone.0173708"

5. The manuscript is too brief on Western blot analyses and comparing to published reports from the SLN field. Please address the following topics.

5a. Mouse quadriceps were subjected to histology staining and force measurements, but not Western blotting for SLN. To the reviewers knowledge, SLN expression has not been reported in mouse quadriceps. To support the claim that reducing SLN expression enhances contractility and myocyte viability in quadriceps of DKO:SLN+/- and TKO mice, please add SLN Western blots of mouse quadriceps to the manuscript.

5b. The range and amount of SLN protein expression in mouse muscles is incompletely known. To the reviewers knowledge, SLN protein expression in mouse has been previously detected at significant level only in soleus and diaphragm. A trace level of SLN has also been detected in mouse EDL, red gastrocnemius (superficial calf), and tibialis anterior (TA). The manuscript should discuss their novel finding of SLN expression in mouse pectoralis.

5c. Even less is known about the expression of SLN in healthy human muscles; if SLN is not expressed in important human muscles (eg, quads, diaphragm, ventricle, etc), then SLN therapy may not be beneficial for DMD patients. Does the manuscript report the first measurement of SLN protein in human DMD muscle? If yes, please add discussion on the significance of their finding (SLN protein is increased 55% in the quadriceps of two DMD patients). The manuscript should also discuss the caveat that SLN gene therapy will depend on the expression pattern of SLN in human tissues.

5d. Trace levels of SLN have been detected in mouse ventricle (Babu PNAS 2007, Witayavanitkul AJP 2013), but in other reports, SLN expression was undetectable in mouse ventricle (Vangheluwe Biochem J 2005, Bal Nat Med 2012). The manuscript should discuss this discrepancy, with respect to its current findings (Fig S1 Western blot: SLN detected in one mouse ventricle, but not in a second sample). The 30-fold upregulation of SLN in mouse DKO ventricle (reported in the manuscript) is reminiscent of Takotsubo cardiomyopathy, whereby SLN expression is upregulated many-fold in the ventricles of patients for 2-4 weeks post-attack. To strengthen the claim that SLN is a therapeutic target in human DMD and HF, please add discussion and citation to the SLN-Takotsubo article: Abnormalities in intracellular Ca²⁺ regulation contribute to the pathomechanism of Tako-Tsubo cardiomyopathy (Nef, Eur Heart J. 2009).

5e. The relative amount of SLN protein expression should be compared directly between the five mouse muscles and reported in the manuscript. This is important information that is missing from the literature and the current manuscript. There are a few conflicting results reported to date: SLN expression in mouse atria is >100-fold higher than soleus (Vangheluwe, Biochem J 2005), SLN expression in mouse atria is 2-fold lower than soleus and diaphragm (Shaikh, J Mol Cell Cardiol 2016), and SLN expression in mouse soleus is 80-fold lower than SERCA expression (Butler, Arch Biochem Biophys 2015). The manuscript should correlate SLN expression in the five mouse muscles with results obtained from the main assays.

6. Three topics to address on Ca uptake assays (7a-c).

6a. All graphs of Ca uptake vs Ca concentration have an incorrectly-labeled x-axis (Figs 1C, S1C, S1F). These graphs are linear-log plots of Ca activation, yet the x-axis jumps from $pCa = 8$ to $pCa = 6$ and 5 in 1 log unit (ie, 100 nM to 1 and 10 mM, on an unbroken axis).

6b. All Ca activation curves of Ca transport in manuscript are not analyzed or reported quantitatively (Figs 1C, S1C, S1F). Manuscript text describes SERCA activity as "higher" for certain mouse strains, without reporting V_{max} . Ca activation curves should be fitted to Hill equation, and appropriate enzyme parameters should be reported (V_{max} , KCa , nH). This is particularly important since SLN has been reported to (i) inhibit or activate V_{max} , (ii) inhibit or have no effect on KCa , and (iii) uncouple Ca transport from ATPase hydrolysis, whereby SERCA 'coupling ratio' is less than the optimal ratio of 2 Ca ions transported per ATP molecule hydrolyzed, due to uncoupling by SLN). If SLN control of SERCA activity is indeed responsible for DMD mitigation effects reported in the manuscript, then SERCA function and SLN inhibitory effects must be reported quantitatively.

6c. The manuscript claims that SERCA activity is inversely correlated to SLN expression levels (fourth paragraph on the first page after the Abstract). Please demonstrate this unsupported claim (graph or text).

7. The manuscript reports small sample sizes: typically $n=3-5$ per group in the six main assays.

Minor comments

7a. The manuscript contains impressive histology and includes a scale bar (without scale size) in every histology Fig. To help the reader, please add the size of scale bar above the scale bar in each Fig (ie, instead of scale description buried in the text of long Fig captions), at least once per Fig panel. There is a lot of histology in manuscript (Figs 2D, 2E, 3F, S3A, S3B, S5C) with a mix of techniques (H&E, trichrome, WGA, immunostaining), and not all assayed are applied to all samples. Fig S3 labels A = 'diaphragm' and B = 'quadriceps' on the left side of the histology panels, which is useful information; perhaps the manuscript would like to add 'ventricle' label to left side of Fig 3 histology in main manuscript. Identifying species directly in Figs would also be useful (eg, add text labels for dog and human DMD samples), or at least, identify the species in the title of Fig legends.

7b. Is the histology scale bar in Fig 3F correct? It looks to be 2-fold too long (ie, appropriate for 20X objective and 100 μm scale, as in Fig S3A), instead of the reported 10X objective with 100 μm scale bar. Please compare scale bars of mouse ventricle histology in Fig 3F vs. Fig S5C, which are 2-fold different in scale length, yet the tissue cellular architecture looks similarly sized in both Figs.

7c. Figure 2E indicates that the pectorals sections were immunostained for myosin heavy chain 1 and embryonic isoforms (Fig panel left labels), yet the legend to Figure 2E indicates immunostaining for MHC1 and SERCA2a. Please fix: SERCA2a or eMHC?

8. In general, the manuscript uses multiple, mixed terms for mice strains (eg, DKO and $mdx/utr-:utr-$; tKO and $mdx/utr-:utr-/sln-:sln-$). As another example, Figs S1 and S2 use three terms for muscles: WT (mice), N (normal; dogs), and Non-DMD (human). Please make terminology and abbreviations more uniform, for ease of the reader.

9. Preparation of tissue homogenates is not described in manuscript. This is particularly important because of the susceptibility of SERCA to proteolysis. For instance, the Western blot in Fig S1D shows that (a) SERCA1a is 25-100% proteolyzed in 4 of 6 dog ECU samples and (b) SERCA2a is ~25% proteolyzed in 3 of 6 dog ECU samples. In addition, the Western blot in Fig S2A shows proteolysis of SERCA1a in 3 of 4 human quadriceps samples. The manuscript should address the issue of SERCA proteolysis, and discuss how it would affect activity measurements (eg, dog WT and DMD samples reported in Fig S1F).

10. Western blot results of WT and DMD dogs (Fig S1D-F) are described by 1 sentence in manuscript (12 words), yet this avenue of investigation appears to have novel results, as reported

in Supplementary Fig. S1; that is, in ECU muscle, SERCA2a/SERCA1a expression pattern switches in normal vs DMD dogs, with 3-fold upregulation of SERCA1a and 2-fold downregulation of SERCA2a in DMD ECU (Fig S1D,E). Activity measurements further show that SERCA activity is decreased in DMD muscle (Fig S1F). Are these Western and activity assays novel results? If yes, please discuss in manuscript. If not, either (a) add discussion of results to manuscript with appropriate citation, or (b) remove this data from Supplement because it adds unnecessary/unreferenced material to the manuscript (ie, information bloat for the reader).

11. In Methods, please describe the CSQ antibody selectivity: skeletal CSQ1 and/or heart CSQ2?

12. Methods section, SR Ca uptake: change μm (micron) to μM (micromolar).

13. Please add a sentence and citation describing the tropism of AAV9. (Several authors are indeed world experts in AAV gene therapy: exciting work!)

14. The manuscript reports a scattered, yet large, collection of assays on muscles from different species; for example, Western blot of 6 tissues, histology of 4 tissues, Ca uptake on 3 tissues, force measurement on 2 tissues, and calpain assay on 1 tissue. Please help the reader understand assay selection and tissue prioritization; for example, why was the calpain assay run only on mouse pectoralis? Two other examples with limited justification are (a) mouse quadriceps, which was subjected to histology, Ca uptake, and force measurement assays, but not calpain assay or Western blot, and (b) Ca uptake was performed on homogenates for 3 of 7 muscles examined in the manuscript: mouse diaphragm, mouse ventricle, and dog ECU. The manuscript identifies two severely affected muscles (mouse pectorals and mouse diaphragm) and one less-severely affected muscle (mouse EDL). Please add additional rationale for tissue/assay selection. Perhaps a Table or list of tissue vs. assay would be informative and help summarize the overall body of work in the manuscript for the reader.

Suggestions for Figures and Tables

1. Fig 1A,B bar graphs: remove 'in' from y-axis units "Body weight (in gm)"

2. Fig 1D,E bar graphs: change y-axis label 'Fold changes' to 'Protein expression'

3. Fig 1C graph: change y-axis label "Percent survival" to "Survival (%)"

4. Figs 2B,C, 3F, and 4C,D bar graphs: there seems to be an extra space in y-axis units "Area (%_)"

5. Fig S1B bar graph: change y-axis label 'SLN levels' to 'SLN expression'

6. Fig 3B, 3C, 4G, 4H bar graphs: change y-axis label "Normalized force (N/g)/2 Hz" to "Force (N/g at 2 Hz)"

7. Figs 1C, S1C, S1F activity graphs: add temp to y-axis units '(nmol/mg/min)' to '(nmol/mg/min at 37C)'

8. Fig S1E bar graph: change y-axis label 'Fold changes' to 'Protein expression'

9. Fig S2B bar graph: change y-axis label 'Fold changes' to 'Protein expression'

10. Fig S2D bar graph: change y-axis label 'SLN fold changes' to 'SLN expression'

11. Fig S5A bar graph: change y-axis label 'Fold changes' to 'Protein expression'

12. Fig S5B bar graph: change y-axis label 'SLN fold changes' to 'SLN expression'

13. Methods section, statistical analysis: change "P" value to "p" value.

14. Supplementary Table 1 uses two symbols (\$ and ++) for the same p value ($p < 0.05$ vs. WT).

15. Supplementary Table reports that LVIDs of TKO mice show $p = 0.06$ vs WT and DKO mice (which likely cannot be correct, due to different LVIDs values of WT and KDO mice). Was "=" used instead of "<"?

Signed ,

David D. Thomas and Joseph M. Autry
University of Minnesota

Responses to the reviewers' critiques

We are grateful to the editor and the reviewers for recognizing the significance of this study and for their helpful and constructive critiques and suggestions. We have revised the manuscript by taking into account each point raised by the reviewers. These changes are highlighted in “red” color fonts in the revised manuscript. The manuscript is now formatted to the requirements of “Nature communications”.

Point by point responses to the reviewers' comments are listed below:

Reviewer # 1

1) This paper does not extend our mechanistic understanding of DMD. For example, why does dystrophin deficiency increase SLN?

We agree with the reviewer that the current study is focused to determine the beneficial effects of reducing SLN expression in mitigating DMD. Although SLN upregulation is an early and intrinsic molecular change to dystrophin deficiency (please see the data provided for Reviewer #2 in page# 5), at present, we do not know how dystrophin deficiency increases SLN expression. Our future studies will determine the molecular mechanisms associated with SLN activation in dystrophin deficient muscles.

How does reducing SLN and increasing SERCA activity in the muscle fibers decrease fibrosis?

In muscles, SERCA accounts for >70% Ca^{2+} removal from the cytosol. We therefore hypothesized that increased SLN levels could chronically inhibit SERCA pump and contribute for the sustained elevation of cytoplasmic Ca^{2+} concentration in dystrophic muscles. The raise in cytoplasmic Ca^{2+} could activate calpain, a Ca^{2+} dependent protease which is shown to play a pivotal role in tissue remodeling and fibrosis. Accordingly, our studies show that improving SERCA function via reducing SLN expression prevents calpain activation and tissue remodeling. This point is now discussed in the revised manuscript (Discussion, page # 9, 1st paragraph).

2) The lifespan extension of the heterozygous SLN deficient *mdx/utr*^{-/-} mice is very impressive. What does a deficiency in SLN do to WT mice? For example, how does SLN deficiency affect lifespan in the presence of dystrophin (Fig. 1b)? What are the consequences to both the *mdx/utr*^{-/-} mice and dystrophin normal mice with exercise? These issues should at least be discussed.

We have previously reported that SLN deficient mice (*sln*^{-/-}) are normal as evident by no change in body weight, survival and quality of life (Babu et al, 2007, PNAS). On the other hand, these mice show improved atrial and skeletal muscle function. These points are now discussed in the revised manuscript (Discussion, page # 11).

Based on the literature, muscle pathology could be worsened in *mdx:utr*^{-/-} mice by forced exercise compared to WT (dystrophin normal) mice. Our future studies will explore whether reduction in SLN expression can prevent these contraction induced muscle damage in *mdx:utr*^{-/-} mice

3)What is the significance of the increased CSQ levels in *mdx/utr*^{-/-} mice?

Re-expression of CSQ along with SERCA2a in dystrophic fast-twitch muscles (like quadriceps and pectoral muscles) may be a compensatory alteration to prevent deficiencies in SERCA function. Improving SERCA function upon SLN reduction revert these changes supports this hypothesis. This point is now discussed in the revised manuscript (Discussion, page #11, 1st paragraph).

4)The *mdx/utr*^{-/-}/*sln*^{-/-} show improved life span but the histology does not appear to be greatly improved in supplemental Fig.3. Does this mean that most of the *mdx/utr*^{-/-} mice normally die of heart failure?

The low ejection fraction (EF) in the *mdx:utr*^{-/-} mice is a sign of heart failure. In the *mdx:utr*^{-/-}:*sln*^{+/-} and tKO mice, the EF numbers are similar to that of WT mice indicating improved cardiac function in these mice.

The beneficial effects of SLN deficiency seem to be greater in the heart than in skeletal muscle. This should be discussed.

We agree with the reviewer that the beneficial effect of total SLN loss is more pronounced in the heart than skeletal muscles in the *mdx:utr*^{-/-} mice. As suggested by the reviewer, this point is now discussed as below in page #11.

“Either reduction or total loss of SLN equally improved the LV function and reduced the myocardial fibrosis and necrosis in the *mdx:utr*^{-/-} mice. These findings suggest that the beneficial effect of total SLN loss is more pronounced in the heart than skeletal muscles in the *mdx:utr*^{-/-} mice. Therefore, it is possible that the improved cardiac function can partly contribute to the improvement in quality of life and survival of both *mdx:utr*^{-/-}:*sln*^{+/-} and tKO mice.”

Based on Supplemental Figure 3, it is difficult to see how the data in fig 2b and c with respect to necrosis and fibrosis were obtained. Perhaps the authors need to provide a more representative figure.

As suggested by the reviewer, we now provided new representative images (Figure 2b) for H& E and Masson’s trichrome-stained quadriceps and diaphragm.

5)Likewise the panel for eMyHC for the *mdx/utr*^{-/-} mice does not appear to be very representative of the summarized data (looks a lot more than 40%).

The eMyHC or Type I MyHC expressing muscle fibers were grouped and not evenly distributed in dystrophic muscles of *mdx:utr*^{-/-} mice. Thus, the data appear more than 40% compared to the quantitation data from the entire tissue sections.

The new representative images showing eMyHC or Type I MyHC expressing fibers are closely matches with the quantitation (new Fig. 3b).

6)In figure 3 is the wt grip strength significantly different for the *mdx/utr*? An n of 7 is very low for grip strength measurements. Are the error bars SEM or SD?

We have increased the *n* numbers for grip strength measurement (please see new Fig. 4a; *n*=9 for *mdx:utr*^{-/-} and *n*=10 for other groups). The WT grip strength values are significantly different from the *mdx:utr*^{-/-} mice (*p*< 0.0001). The error bars represent “mean ±SEM”. This information is included both in the figure and in the figure legend.

7)The force frequency curves in Fig. 3d and e do not look like normal curves for these muscles. The force should be normalized to cross-sectional area rather than muscle weight. Second, the stimulation frequency that produces half maximal force appears to be somewhat low for both EDL and diaphragm from 3-4 month old mice. The data should be fit, maximal force (normalized to CSA), and the half maximal stimulation frequency should be provided.

As suggested by the reviewer we have modified the data analysis. The new figures 4d and 4g show the curve fit for percent maximum force (normalized to CSA) generated by EDL and diaphragm respectively. The half maximal stimulation frequency for each mouse group is given in the results section (Page #6, last paragraph).

Also, the WT and *mdx:utr*^{-/-} data in Fig3a and Supplemental figure 6a appear to be the same.

Although it appears to be the same, figure 3a and Supplementary Fig 6a were derived from two entirely different experiments. Figure 3a was a grip-strength data and figure 6a was a rising slope data for EDL. With the addition of more *n* to grip-strength studies, the new grip strength data (Fig. 4a) looks different from the Supplementary fig. 6a.

Shouldn't the data in Sfig 6 for AAV-shSLN mice be compared to mice receiving an empty AAV9 vector and also use littermate mice?

We agree with the reviewer. The coauthors, Dr. Roger Hajjar and Dr. Dongsheng Duan have been involved with AAV studies and their laboratories have utilized a variety of control vectors including adeno associated virus like particles (AAV.VLP). According to their findings, there have been no effects of control or AAV.VLP on disease progression in a variety of CV models and in dystrophic mice and dogs [Suckau et al (2009), *Circulation* 119: 1241-1252; Hadri et al (2013), *Circulation* 128: 512-523; Sakata et al (2006) *Mol. Ther* 13: 987-996) Ghosh et al (2009), *Human Gene Ther* 20: 1319-1328; Liu et al (2005), *Mol. Ther* 11: 245-256; Shin et al (2012), *Human Gene Ther* 23: 202-209; Yue (2015), *Hum Mol Genet* 24:5880-5890]. In addition, the FDA requires the control arm to be saline injection for gene therapy pre-clinical trials which prompted us to use saline injection as control. Furthermore, due to the limited availability and survival of *mdx:utr*^{-/-} mice, we were unable to use the empty AAV9 vector injected *mdx:utr*^{-/-} controls in our studies.

8)The authors need to show representative echocardiograms for each type of mouse rather than just summarized data.

As suggested by the reviewer, we now provided representative echo images (Please see the supplementary figure 4b).

9) Fig. 3 b and c and Supplementary Fig 6 should show representative twitch force tracings illustrating the measurements (rising slope and decay slope) summarized in the bar graphs.

As suggested by the reviewer, we now provided representative twitch force traces for EDL and diaphragm at 2 Hz. Please refer to figure 4b and 4e.

10) Supplementary fig 2 (the human data) has only an n of 2 seems premature and should be eliminated.

We were unable to obtain more human DMD tissues for this study. Although *n* of 2 is not sufficient to show the statistical significance, our study is the first report demonstrating SLN protein upregulation in both skeletal and ventricles of human DMD. Given the importance of this data in providing strong rationale for mouse model studies, we would like to keep the data at least in the supplementary section.

11) the authors need to state in the figure legends if the data are mean +/- SEM or SD and state the statistical test used for that particular data set.

Data are presented as mean \pm SEM. The statistical test used is given in the figure legend.

Reviewer #2

1. Statistical power

The study on animal models needs to be refined. Although largely consistent, the in vivo data are missing 'n'. The authors looked at cardiac function on 'n'=4. This is largely insufficient for this type of study. Given the non-invasive methods performed in this study (grip strength, echocardiography, survival), one could wonder why not all mice have been assessed for all the studied parameters. This is crucial.

Some of the assays were done with minimal number of animals for the following reasons: 1) Not all animals were used for all studies due to differences in tissue preparation for biochemistry, histology and immunostaining and muscle mechanics and animals used for echocardiography 2) Due to the complexity of genotype, breeding and maintaining the *mdx:utr*^{-/-} and tKO mice were very difficult and challenging. The experimental mice were generated by crossing the male and female *mdx:utr*^{+/-}:*sln*^{+/-} mice. The number of *mdx:utr*^{-/-} and tKO pups born from these breeders were limited (anywhere from 0 to 2 per breeding) and 3) Due to early and sudden death of *mdx:utr*^{-/-} mice.

However, to increase the *n* number, we have repeated some of the studies. The data are consistent with our previous studies. Accordingly, we have updated the following studies with a larger "n".

- 1) Figure 1b- Survival graph
- 2) Figure 1f- New western data on quadriceps, *n*=4
- 3) Figure 2a, calpain assay, *n*=5
- 4) Figure 5b and 5c, and Table 1, Echocardiography *n*=6

- 5) Supplementary figure 1a & 1b, SLN western -ventricles of *mdx* and *mdx:utr*^{-/-} mice, *n*=6
- 6) Figure 4a-Grip strength measurements (*n*=9 for *mdx:utr*^{-/-} and *n*=10 for other groups).
- 7) Figure 4c-4g- Force measurements in isolated EDL and diaphragm.

2. SLN expression should be assessed along the progression of the disease in the animal models. It is important to understand the pathogenic role of sarcolipin in a pre- and post-sympatomatic stage.

Sarcolipin levels in one-month and 4 month-old *mdx:utr*^{-/-} mice tissues are the same, indicating that SLN upregulation is not altered upon progression of the disease in animal models. *[Redacted]*

[Redacted]

1. the manuscript needs a deep english correction

The revised manuscript was read and edited by the native English speaking co-authors.

2. FigS1, why the authors show 'n'=2 on immunoblots and 'n'=3 on quantification. I found it rather odd. They could have showed all the samples on their SDS-PAGE.

We performed additional experiments and increased the *n* numbers (*n*=6). Also we have and provided new western image showing *n*=3 per group. Please see the supplementary figures 1a and 1b.

3. Each figure should be on one and only page.

As per the reviewer's suggestion, each figure was compiled to fit one page.

4. the histology staining should be homogenised. The authors should show a lower magnification of the tissue with an inset of their magnification (especially for heart sections).

We have revised the histology images to show clear differences between the genotypes. For quadriceps and ventricles of AAV and saline treated groups, we have also provided low

magnified (4X) images with 40X inset showing the necrotic areas with mononuclear invasion (please see the supplementary figure # 5).

5. Material Method section is poorly documented. There are no data on dog GRMD samples, the age of the animals are NEVER documented.

“Methods” section is updated with more information about the dog model of DMD as well as human DMD samples. Please see page # 14

Reviewer #3

1. Organization: Here we point out problems in organization, but leave it to the Editors to determine whether these are consistent with the standards and length constraints of the journal. The manuscript is 6 page of text, including ‘Introductory Paragraph’. There are no section headers or subsections for Results or Discussion. There is a Methods section (5 pages) added at the end of the manuscript; is Methods intended as Supplementary Information? There are two separate reference sections: Manuscript (‘Literature Cited’) and Methods (‘References’). Pages of the manuscript are not numbered. The Figure legends are long. In the four Figs of the main manuscript, the average length of caption is 190 ± 32 words, ie, all are longer than the Introductory Paragraph (ie, Abstract, which is 129 words). Much of the text in Fig captions may be more appropriate for inclusion in Results or Methods sections. The captions in Supplement are 105 ± 49 words for 6 SI Figs. The main manuscript is short in length because many results are described in Fig captions.

The manuscript is now formatted to the requirements of “Nature Communications-Article” by including an abstract and separate results and discussion sections. Figures are regrouped. The figure legends are shortened as suggested by the reviewers.

2. More information on is needed on the human muscle samples (quadriceps and heart), per Nat Comm recommended guidelines (‘Biospecimen Reporting for Improved Study Quality’ @ Cancer Cytopathol 119:92–101, 2011). There is no information on human samples (n= 2 for non-DMD and DMD) in the Methods sections. Please describe how human biopsy samples were harvested, processed, and stored for Western blots (eg, location of biopsies in heart: ventricle or atrium?). Other information which needs to be added are selection process for donors and clinical characterization of DMD patients and healthy volunteers.

Details included in the “Methods” section as suggested. Please see page # 14.

3. More information is needed on the dog DMD samples (ECU), per Nature Comm recommended guidelines (‘Animals in Research: Reporting In Vivo Experiments’ @ PLoS Biol. 8(6):e1000412, 2010). There is no information on dog samples (n= 2 for normal and DMD) in the Methods sections. Please add a sentence and citation for how the dog DMD model was generated. Please describe how dog quadriceps were harvested, processed, and stored for Western blots and activity assays.

Details included in the “Methods” section as suggested. Please see page # 14.

4. The manuscript is too brief with little discussion of published reports from the SLN field. Please address the following topics.

4a. The hallmark of SLN-KO mice is that they are acutely-sensitive to cold temperature and become obese on a high-fat diet (Periasamy, Nature Med 2012, FASEB J 2013, JBC 2015a,b, JBC 2016). In particular, expression level of SLN in mouse soleus depends on cage temperature (25 or 37C). Please describe the feeding protocol, housing temperature, and running wheel availability for mice strains in the current manuscript. Fig 1A demonstrates that reduction and ablation of SLN increases body weight in the DKO background. Please discuss current results in comparison to SLN-KO results from Periasamy lab.

Mice were kept under a 12-hour light/dark cycle with a temperature of 22-24 °C and 60-70% of humidity and fed ad libitum with normal chow diet. This information is now included in the “Methods” section (Page# 13, 2nd paragraph). The normal chow did not affect the body weight of SLN null mice. Results from Dr. Periasamy’s lab were discussed appropriately in the “Discussion” section (Page # 9 &10).

4b. Myoregulin (MLN) has been proposed to be the main regulator of SERCA in mouse muscle (Olson, Cell 2015, Science 2016). According to that paper, SLN protein expression in mouse had been previously detected at significant level only in soleus and diaphragm. The manuscript should discuss the hypothesis from Olson lab that MLN is the main SERCA regulator in mouse muscle.

We have previously reported that SLN is upregulated in the diaphragm, soleus and quadriceps of both *mdx* and *mdx:utr*^{-/-} mice (Schneider et al J Musc. Res. Motil. 2013). Our findings also corroborate with the microarray and semi-quantitative RT-PCR data, which show SLN mRNA upregulation in the medial gastrocnemius of *mdx* mice (Marotta et al, Physiol. Genomics, 2009). These data together suggest that SLN is upregulated in all muscles of DMD mice. As suggested by the reviewer we have included this information along with other SERCA regulators including MLN and discussed appropriately. Please see Discussion, page #9.

4c. A relevant article was published during review which reports that SLN deletion exacerbates muscle disease in a different mouse model (ie, SLN-KO is causative, not compensatory). Please discuss this new article in relation to the current manuscript. “Sarcolipin deletion exacerbates soleus muscle atrophy and weakness in phospholamban overexpressing mice. Fajardo VA, PLoS One. 2017 Mar 9;12(3):e0173708. doi: 10.1371/journal.pone.0173708”

This study is referred and discussed as below: Please refer to discussion, page # 10, 2nd paragraph.

A recent study shows that complete loss of SLN fails to improve SERCA function and centronuclear myopathy like phenotype in transgenic mice with skeletal muscle specific PLN overexpression. On the other hand, SLN ablation resulted in exacerbated muscle atrophy and weakness with impaired calcineurin pathway and suggesting that SLN overexpression may have

compensatory effects on muscle function in these mice. We did not find any possible explanation for this discrepancy with current findings in DMD mice. However, our data also suggest that total loss of SLN is not beneficial to dystrophin deficient skeletal muscles.

5. The manuscript is too brief on Western blot analyses and comparing to published reports from the SLN field. Please address the following topics.

5a. Mouse quadriceps were subjected to histology staining and force measurements, but not Western blotting for SLN. To the reviewers knowledge, SLN expression has not been reported in mouse quadriceps. To support the claim that reducing SLN expression enhances contractility and myocyte viability in quadriceps of DKO:SLN^{+/-} and TKO mice, please add SLN Western blots of mouse quadriceps to the manuscript.

We have previously reported that SLN is upregulated in the diaphragm, soleus and quadriceps of both *mdx* and *mdx:utr*^{-/-} mice (Schneider et al J Musc. Res. Motil. 2013). As suggested by the reviewer we have now included western data on SLN, SERCA isoforms and CSQ expression in quadriceps muscles. Results shown in Fig. 1f corroborate with our previous results on the upregulation of SLN, SERCA2a and CSQ in the quadriceps of *mdx:utr*^{-/-}. Furthermore, reduction or ablation of SLN restored the SERCA isoform expression as well as normalized CSQ levels in the quadriceps of *mdx:utr*^{-/-} mice.

5b. The range and amount of SLN protein expression in mouse muscles is incompletely known. To the reviewers knowledge, SLN protein expression in mouse has been previously detected at significant level only in soleus and diaphragm. A trace level of SLN has also been detected in mouse EDL, red gastrocnemius (superficial calf), and tibialis anterior (TA). The manuscript should discuss their novel finding of SLN expression in mouse pectoralis.

Findings from the present study along with our earlier report (Schneider et al J Musc. Res. Motil. 2013) have demonstrated that in DMD mice, SLN but not PLN is elevated in the diaphragm, and slow- (soleus) and fast-twitch skeletal muscles including quadriceps and pectoral muscles as well in atria and in the ventricles. Our findings also corroborate with the microarray and semi-quantitative RT-PCR data, which show SLN mRNA upregulation in the medial gastrocnemius of *mdx* mice (Marotta et al, Physiol. Genomics, 2009). Together these findings suggest that SLN upregulation is a common molecular change in all skeletal muscle tissues and in the heart in DMD. This point is now discussed in page #9, 2nd paragraph.

5c. Even less is known about the expression of SLN in healthy human muscles; if SLN is not expressed in important human muscles (eg, quads, diaphragm, ventricle, etc), then SLN therapy may not be beneficial for DMD patients. Does the manuscript report the first measurement of SLN protein in human DMD muscle? If yes, please add discussion on the significance of their finding (SLN protein is increased 55% in the quadriceps of two DMD patients). The manuscript should also discuss the caveat that SLN gene therapy will depend on the expression pattern of SLN in human tissues.

SLN is expressed differently in rodents and in larger mammals. In rodents, SLN is predominantly expressed in the tongue followed by diaphragm and slow-skeletal muscles but not

expressed in the fast-twitch skeletal muscles (Babu et al JMCC, 2007 and Anderson et al, Sci Signal, 2016). In contrast, in larger mammals SLN expressed in all skeletal muscle tissues (Babu et al JMCC, 2007). In the heart, PLN expression is high in the ventricles. On the other hand, SLN expression is very high in atria and very low in the ventricles of both rodents and larger mammals (Babu et al JMCC, 2007). Our unpublished data also show that SLN is expressed at high levels in normal human skeletal muscles. The current study along with gene expression analyses, which show increased levels of SLN mRNA in many of the biopsy samples obtained from DMD patients (NIH databaseGDS214 /HSPD1705_g_at/SLN/Homo sapiens, <http://www.ncbi.nlm.nih.gov/geo/profiles/999633>) suggest that SLN upregulation could be a common molecular change in all skeletal muscles and in the heart in DMD patients. Thus the whole body SLN gene therapy may be beneficial in DMD patients.

5d. Trace levels of SLN have been detected in mouse ventricle (Babu PNAS 2007, Witayavanitkul AJP 2013), but in other reports, SLN expression was undetectable in mouse ventricle (Vangheluwe Biochem J 2005, Bal Nat Med 2012). The manuscript should discuss this discrepancy, with respect to its current findings (Fig S1 Western blot: SLN detected in one mouse ventricle, but not in a second sample).

The band seen in Fig S1 was a non-specific band, appears slightly above the SLN. Now, we have repeated these studies with new set of samples and presented in supplementary figure 1a.

For the discrepancy in findings of SLN levels in mouse ventricles, please see the responses to question 5e.

5e. The relative amount of SLN protein expression should be compared directly between the five mouse muscles and reported in the manuscript. This is important information that is missing from the literature and the current manuscript. There are a few conflicting results reported to date: SLN expression in mouse atria is >100-fold higher than soleus (Vangheluwe, Biochem J 2005), SLN expression in mouse atria is 2-fold lower than soleus and diaphragm (Shaikh, J Mol Cell Cardiol 2016), and SLN expression in mouse soleus is 80-fold lower than SERCA expression (Butler, Arch Biochem Biophys 2015). The manuscript should correlate SLN expression in the five mouse muscles with results obtained from the main assays.

The discrepancy seen in different studies on the SLN protein levels in the ventricles is due to the specificity of the SLN antibody used. We have previously generated a rabbit polyclonal antibody specific for the C-terminal region of SLN. This antibody recognizes SLN across species (including human) with a same sensitivity (Babu et al JMCC 2007). We require only few micrograms of total proteins to detect SLN in the atria of rodents (Babu et al JMCC 2007), compared to other studies which used 50 to 100 microgram proteins to detect SLN in atria. Based on the literature and current findings, SLN upregulation appears to be a common molecular change in all muscles and in the heart (both atria and in the ventricles). This point is now discussed appropriately (Page# 9, 2nd paragraph).

The 30-fold upregulation of SLN in mouse DKO ventricle (reported in the manuscript) is reminiscent of Takotsubo cardiomyopathy, whereby SLN expression is upregulated many-fold in

the ventricles of patients for 2-4 weeks post-attack. To strengthen the claim that SLN is a therapeutic target in human DMD and HF, please add discussion and citation to the SLN-Takotsubo article: Abnormalities in intracellular Ca²⁺ regulation contribute to the pathomechanism of Tako-Tsubo cardiomyopathy (Nef, Eur Heart J. 2009).

In collaboration with Dr. Lou Dell'Italia at UAB, we have previously reported that SLN protein levels were significantly increased in the ventricles of patients with mitral regurgitation (Zheng et al, Circ Heart Fail. 2014). These studies along with the RNA data published by Nef et al (2009) in Tako-tsubo cardiomyopathy suggest that SLN upregulation may have similar role in the cardiac pathogenesis during volume overloaded hypertrophy and Tako-tsubo cardiomyopathy. SLN could be a novel therapeutic target, if the observed beneficial effects of SLN downregulation in DMD-cardiomyopathy can be recapitulated in these diseases. This point is now discussed (page#10, last paragraph).

6. Three topics to address on Ca uptake assays (7a-c).

6a. All graphs of Ca uptake vs Ca concentration have an incorrectly-labeled x-axis (Figs 1C, S1C, S1F). These graphs are linear-log plots of Ca activation, yet the x-axis jumps from pCa = 8 to pCa = 6 and 5 in 1 log unit (ie, 100 nM to 1 and 10 mM, on an unbroken axis).

The images shown in the previous submission were automatically generated by the GraphPad Prism software. We have now provided a new Ca²⁺ uptake graph with corrected “x” axis. Please see figure 1c and supplementary figure 1c and 1f.

6b. All Ca activation curves of Ca transport in manuscript are not analyzed or reported quantitatively (Figs 1C, S1C, S1F). Manuscript text describes SERCA activity as “higher” for certain mouse strains, without reporting V_{max}. Ca activation curves should be fitted to Hill equation, and appropriate enzyme parameters should be reported (V_{max}, K_{Ca}, nH). This is particularly important since SLN has been reported to (i) inhibit or activate V_{max}, (ii) inhibit or have no effect on K_{Ca}, and (iii) uncouple Ca transport from ATPase hydrolysis, whereby SERCA ‘coupling ratio’ is less than the optimal ratio of 2 Ca ions transported per ATP molecule hydrolyzed, due to uncoupling by SLN). If SLN control of SERCA activity is indeed responsible for DMD mitigation effects reported in the manuscript, then SERCA function and SLN inhibitory effects must be reported quantitatively.

Thank you for pointing out the missing data. We have now provided the EC₅₀ values and the maximum velocity of Ca²⁺ uptake (V_{max}) for each group. For details, please see the “results” section in page # 4 and 5. The increased V_{max} of Ca²⁺ uptake and the increased apparent binding affinity of SERCA pump for Ca²⁺ (as shown by decreased EC₅₀) in the *mdx:utr^{-/-}:sln^{+/-}* and tKO muscles also suggests that SLN upregulation could be a major cause of SERCA dysfunction and elevation of Ca²⁺_i in DMD. This point is now discussed in page #10, 1st paragraph.

6c. The manuscript claims that SERCA activity is inversely correlated to SLN expression levels (fourth paragraph on the first page after the Abstract). Please demonstrate this unsupported claim (graph or text).

We have modified this statement as there is no direct evidence.

7. The manuscript reports small sample sizes: typically $n=3-5$ per group in the six main assays.

Some of the assays were done with minimal number of animals for the following reasons: 1) Not all animals were used for all studies due to differences in tissue preparation for biochemistry, histology and immunostaining and muscle mechanics and animals used for echocardiography 2) Due to the complexity of genotype, breeding and maintaining the *mdx:utr*^{-/-} and tKO mice were very difficult and challenging. The experimental mice were generated by crossing the male and female *mdx:utr*^{+/-};*sln*^{+/-} mice. The number of *mdx:utr*^{-/-} and tKO pups born from these breeders were limited (anywhere from 0 to 2 per breeding) and 3) Due to early and sudden death of *mdx:utr*^{-/-} mice.

However, to increase the n number, we have repeated some of the studies. The data are consistent with our previous studies. Accordingly, we have updated the following studies with a larger “ n ”.

- 1) Figure 1b- Survival graph
- 2) Figure 1f- New western data on quadriceps, $n=4$
- 3) Figure 2a, calpain assay, $n=5$
- 4) Figure 5b and 5c, and Table 1, Echocardiography $n=6$
- 5) Supplementary figure 1a & 1b, SLN western -ventricles of *mdx* and *mdx:utr*^{-/-} mice, $n=6$
- 6) Figure 4a-Grip strength measurements ($n=9$ for *mdx:utr*^{-/-} and $n=10$ for other groups).
- 7) Figure 4c-4g- Force measurements in isolated EDL and diaphragm.

Minor comments

7a. The manuscript contains impressive histology and includes a scale bar (without scale size) in every histology Fig. To help the reader, please add the size of scale bar above the scale bar in each Fig (ie, instead of scale description buried in the text of long Fig captions), at least once per Fig panel. There is a lot of histology in manuscript (Figs 2D, 2E, 3F, S3A, S3B, S5C) with a mix of techniques (H&E, trichome, WGA, immunostaining), and not all assayed are applied to all samples. Fig S3 labels A = ‘diaphragm’ and B = ‘quadriceps’ on the left side of the histology panels, which is useful information; perhaps the manuscript would like to add ‘ventricle’ label to left side of Fig 3 histology in main manuscript. Identifying species directly in Figs would also be useful (eg, add text labels for dog and human DMD samples), or at least, identify the species in the title of Fig legends.

As suggested by the reviewer, we have now labelled all the images appropriately with the name of the tissue and species. Also, we have added the size of the scale bar in the figure itself.

7b. Is the histology scale bar in Fig 3F correct? It looks to be 2-fold too long (ie, appropriate for 20X objective and 100 μ m scale, as in Fig S3A), instead of the reported 10X objective with 100 μ m scale bar. Please compare scale bars of mouse ventricle histology in Fig 3F vs. Fig S5C, which are 2-fold different in scale length, yet the tissue cellular architecture looks similarly sized in both Figs.

Thank you for pointing out the mistake, which is corrected in the text.

7c. Figure 2E indicates that the pectoral muscle sections were immunostained for myosin heavy chain 1 and embryonic isoforms (Fig panel left labels), yet the legend to Figure 2E indicates immunostaining for MHC1 and SERCA2a. Please fix: SERCA2a or eMHC?

We now provide new images representing fibers expressing either eMyHC or Type I MyHC. The figure legend is also modified accordingly.

8. In general, the manuscript uses multiple, mixed terms for mice strains (eg, DKO and *mdx/utr-:utr-*; tKO and *mdx/utr-:utr-/sln-:sln-*). As another example, Figs S1 and S2 use three terms for muscles: WT (mice), N (normal; dogs), and Non-DMD (human). Please make terminology and abbreviations more uniform, for ease of the reader.

We now used uniform abbreviations throughout the manuscript. Specifically, *mdx:utr:sln-/-* mouse is abbreviated as triple knockout (tKO), normal dogs are relabeled as Non-DMD as in human.

9. Preparation of tissue homogenates is not described in manuscript. This is particularly important because of the susceptibility of SERCA to proteolysis. For instance, the Western blot in Fig S1D shows that (a) SERCA1a is 25-100% proteolyzed in 4 of 6 dog ECU samples and (b) SERCA2a is ~25% proteolyzed in 3 of 6 dog ECU samples. In addition, the Western blot in Fig S2A shows proteolysis of SERCA1a in 3 of 4 human quadriceps samples. The manuscript should address the issue of SERCA proteolysis, and discuss how it would affect activity measurements (eg, dog WT and DMD samples reported in Fig S1F).

Details of tissue storage and protein preparations are now included in the “Methods” section. At present, we do not know whether the additional band appeared in both Non-DMD and DMD samples (Fig S1d and S2a) are proteolytic fragments of SERCA. Regardless of these changes, the rate and V_{max} of calcium uptake was significantly reduced in the dystrophic muscles.

10. Western blot results of WT and DMD dogs (Fig S1D-F) are described by 1 sentence in manuscript (12 words), yet this avenue of investigation appears to have novel results, as reported in Supplementary Fig. S1; that is, in ECU muscle, SERCA2a/SERCA1a expression pattern switches in normal vs DMD dogs, with 3-fold upregulation of SERCA1a and 2-fold downregulation of SERCA2a in DMD ECU (Fig S1D,E). Activity measurements further show that SERCA activity is decreased in DMD muscle (Fig S1F). Are these Western and activity assays novel results? If yes, please discuss in manuscript. If not, either (a) add discussion of results to manuscript with appropriate citation, or (b) remove this data from Supplement because it adds unnecessary/unreferenced material to the manuscript (ie, information bloat for the reader).

SLN upregulation, altered expression of SERCA isoforms and decreased SR Ca^{2+} uptake in the skeletal muscle of DMD dog are novel and not previously reported. These findings are explained appropriately in the “Results” (page # 4) as well as in the “Discussion” (page #9, 1st paragraph) sections.

11. In Methods, please describe the CSQ antibody selectivity: skeletal CSQ1 and/or heart CSQ2?

According to the manufacturer (Affinity Bioreagents), this antibody should recognize both cardiac and skeletal muscle CSQ. This information is now provided in the “Methods” section.

12. Methods section, SR Ca uptake: change μm (micron) to μM (micromolar).

Thanks for pointing out the mistake. It is now corrected.

13. Please add a sentence and citation describing the tropism of AAV9. (Several authors are indeed world experts in AAV gene therapy: exciting work!)

AAV9 targets both cardiac and skeletal muscles. This information is now included in the discussion.

14. The manuscript reports a scattered, yet large, collection of assays on muscles from different species; for example, Western blot of 6 tissues, histology of 4 tissues, Ca uptake on 3 tissues, force measurement on 2 tissues, and calpain assay on 1 tissue. Please help the reader understand assay selection and tissue prioritization; for example, why was the calpain assay run only on mouse pectoralis? Two other examples with limited justification are (a) mouse quadriceps, which was subjected to histology, Ca uptake, and force measurement assays, but not calpain assay or Western blot, and (b) Ca uptake was performed on homogenates for 3 of 7 muscles examined in the manuscript: mouse diaphragm, mouse ventricle, and dog ECU. The manuscript identifies two severely affected muscles (mouse pectorals and mouse diaphragm) and one less-severely affected muscle (mouse EDL). Please add additional rationale for tissue/assay selection. Perhaps a Table or list of tissue vs. assay would be informative and help summarize the overall body of work in the manuscript for the reader.

We now provided a table showing various tissues and the assays performed in the “Methods” section (page # 13).

Suggestions for Figures and Tables

1. Fig 1A,B bar graphs: remove ‘in’ from y-axis units “Body weight (in gm)”
2. Fig 1D,E bar graphs: change y-axis label ‘Fold changes’ to ‘Protein expression’
3. Fig 1C graph: change y-axis label “Percent survival” to “Survival (%)”
4. Figs 2B,C, 3F, and 4C,D bar graphs: there seems to be an extra space in y-axis units “Area (%_)”
5. Fig S1B bar graph: change y-axis label ‘SLN levels’ to ‘SLN expression’
6. Fig 3B, 3C, 4G, 4H bar graphs: change y-axis label “Normalized force (N/g)/2 Hz” to “Force (N/g at 2 Hz)”
7. Figs 1C, S1C, S1F activity graphs: add temp to y-axis units ‘(nmol/mg/min) to (nmol/mg/min at 37C)’
8. Fig S1E bar graph: change y-axis label ‘Fold changes’ to ‘Protein expression’
9. Fig S2B bar graph: change y-axis label ‘Fold changes’ to ‘Protein expression’

10. Fig S2D bar graph: change y-axis label 'SLN fold changes' to 'SLN expression'
11. Fig S5A bar graph: change y-axis label 'Fold changes' to 'Protein expression'
12. Fig S5B bar graph: change y-axis label 'SLN fold changes' to 'SLN expression'
13. Methods section, statistical analysis: change "P" value to "p" value.
14. Supplementary Table 1 uses two symbols (\$ and ++) for the same p value ($p < 0.05$ vs. WT).
15. Supplementary Table reports that LVIDs of TKO mice show $p = 0.06$ vs WT and DKO mice (which likely cannot be correct, due to different LVIDs values of WT and KDO mice). Was "=" used instead of "<"?

As suggested by the reviewers, we made changes in the figures and tables. Thank you.

REVIEWERS' COMMENTS:

Reviewer #1 (Remarks to the Author):

The authors have adequately addressed my previous concerns

Reviewer #2 (Remarks to the Author):

The authors clearly improved their manuscript by answering some of the comments of the Reviewers. The data of the present manuscript are of interest in the field of muscular dystrophy and therapies. Some crucial points are still missing and could be improved prior publication:

- it seems not perfectly adequate to repeat some experiments using the same samples to reach good statistics.
- The role of SLN in Duchenne muscular dystrophy is clearly missing. This could have been addressed by the authors for a more compelling story.

Reviewer #3 (Remarks to the Author):

Summary

The revised manuscript NCOMMS-17-03326A by Voit, Babu, et al. (Reducing Sarcolipin Expression Mitigates Duchenne Muscular Dystrophy and Associated Cardiomyopathy in Mice) is a tour-de-force of molecular physiology and translational research. The authors performed additional experiments and revised the manuscript per reviewers' comments; as such, the revised manuscript is much improved. Additional major comments (2) and minor comments (10) are provided below.

Major comments

1. The manuscript neglects a key control (Western blot of myoregulin, MLN) needed to interpret results of SERCA functional assays and SLN regulation. The mode of regulation detected in Fig 1C (super-inhibition of SERCA K_{Ca} and V_{max}) is most reminiscent of SERCA super-inhibition by SLN and PLB, as induced by formation of a super-inhibitor ternary complex, where calcium affinity if decreased 5-fold and maximal velocity decreased 25-50% in SERCA-PLB-SLN (MacLennan JBC 2002; see also Tupling PLOS 2013). Olson et al. have proposed that MLN is the key regulator in mouse muscle and that SLN plays only a very minor role (Cell 2015, Sci Signal 2016). For example, MLN is robustly expressed in WT mice in quadriceps, plantaris, gastroc, EDL, TA, diaphragm, and soleus (Cell 2015), whereas SLN was undetected in WT mice in quadriceps, plantaris, gastroc, EDL, and TA. Yes, the current NatComm manuscript does demonstrate a large relative upregulation of SLN protein in 3 mouse muscles (quads, pecs, diaphragm), but the current manuscript does not provide absolute quantitation of SLN. (Is SLN upregulated to the same protein level as SERCA? Does the SLN/SERCA stoichiometry exceed 3-5 to allow for full regulation, as demonstrated in heterologous and reconstitution systems?) The functional data in Fig 1C (WT mice) suggest that SERCA shows high activity in mouse diaphragm (i.e., little-to-no MLN inhibition), yet the experiment needs to be performed to eliminate uncertainty in interpretation: Do mouse diaphragm, pectoralis, and quadriceps express MLN, and how do these levels change in mdx, DKO, and TKO mice? Note: FabGennix now sells the first-commercially available MLN antibody, which is reactive to rodent MLN.

2. Functional effects are typically correlated with expression level of catalytic (SERCA) and regulatory (SLN) subunits. The manuscript does not compare expression levels of SLN or SERCA between mouse tissues, and thus leaves an existing information gap in the literature that the manuscript is poised to finally cover (see Reviewer 3 Comment 5E on the original manuscript submission).

Minor comments

1. Ca uptake is only measured for one mouse muscle tissue: diaphragm, a 'more affected' tissue. Ca uptake is also measured for mouse ventricle and dog ECU, although EC50 is not reported for mouse ventricle (3 mouse lineages). Addition of a Table summarizing Vmax and EC50 for the different tissues and mouse lineages would be helpful for the reader, as would addition of a 'Ca uptake' entry to the new list "Tissues Used" in the Methods section. Perhaps force frequency fits of CSA-normalized force and half-maximal stimulation frequency, per Reviewer 1 Comment 7, could also be added to the Ca uptake table, as the force parameters are currently only listed in text.

2. Ca uptake measurement and analysis: the data are presented as activity vs pCa (Figs 1C, S1C, and S1F), yet the enzyme parameter KCa (Hill fit of Fig 1C graph) is reported as nanomolar (page 4 for S1C, page 5 for Fig 1C). Please pick one unit of Ca concentration (pCa or nM) and use it consistently. Page 4 lines 82-82 incorrectly define EC50 as the 'half maximal velocity of Ca uptake'. The data in Fig 1C are indeed remarkable: ~2.5-fold greater inhibition of SERCA KCa and Vmax in mdx:utr^{-/-} mice (DKO) compared to WT mice, an effect which is removed by knock-out of one SLN allele!

3. SR Ca uptake Methods section on page 17: There appear to be mistakes in the reported amount of ATP and Ruthenium Red added to the assay. 5 mmole of ATP in 1.5 mL = 3.3 M ATP, while 1 umole of RuR in 1.5 mL = 0.67 mM.

4. Fig 1d,e,f. Knockout of one SLN gene allele in DKO mice (mdx:utr^{-/-}) restores SLN expression to normal in diaphragm (similar to WT muscle = almost undetectable SLN expression) and in pectoralis and quadriceps (similar to WT muscle = no SLN expression). SLN protein expression is dramatically upregulated (>10-fold) in all three muscles of DKO mice. Why is there such a dramatic effect upon knockout of 1 SLN allele (i.e., complete reversal of SLN protein expression) instead of a graded 50% reduction upon knockout of 1 SLN gene in sln +/- mice? This is an unexpected result; perhaps it could be corroborated by qRT-PCR, which would quantitate SLN mRNA level (spliced transcripts).

5. Page 9, lines 237-238 has an unsupported claim and citation. "In contrast, in larger mammals SLN is expressed in all skeletal muscle tissues³³." The human body contains hundreds of muscles, so it is unlikely that 'all' have been tested and confirmed to express SLN. There are many types of 'larger mammals', and the SLN global expression pattern in one larger mammal probably does not occur the same for all. The largest animal assayed by SLN Western blot in Ref 33 is rat, with 3 muscles tested: quadriceps, soleus, and diaphragm (Babu et al JMCC 2007), which probably does not serve as an exact model for SLN expression in all human muscles.

6. Page 10, line 409: "mS" Does this abbreviation stand for millisecond or milli Siemen?

7. The manuscript added the number of mice assayed (n) to the middle of each bar in ~half of all the bar graphs in the manuscript (n was added to 17 of 36 bar graphs). Please add 'n' to all bar graphs in manuscript (looks good, useful information, effectively conveyed).

8. There remain small formatting mistakes throughout the manuscript (missing spaces, extra spaces, missing degree symbol, non-superscripted '3' on cm cubed, etc).

9. Proteolytic fragment of SERCA. Reviewer 3, Comment 9. This is probably a proteolytic fragment of SERCA, and not a non-specific immunoreactive band (as suggested in revision letter), since the band is seen on two blots with two different isoform specific antibodies (SERCA1 and 2). For further proof (in the future for the investigators), additional antibodies should be tested (either pan-SERCA or isoform specific), which is easily performed, since dozens of SERCA antibodies are commercially available.

10. The reviewers thank the authors for additional information on their published articles on

quantitation of SLN mRNA and protein expression in various species. Quite complex patterns. It's been 9+ years since a SLN review written by Babu was published, and the pace of the SLN field has been accelerating, so it's probably time for another Babu review on SLN-SERCA.

Signed ,
David D. Thomas and Joseph M. Autry
University of Minnesota

Reviewer 1 comments

The authors have adequately addressed my previous concerns

Thank you.

Reviewer 2 comments

The authors clearly improved their manuscript by answering some of the comments of the Reviewers. The data of the present manuscript are of interest in the field of muscular dystrophy and therapies. Some crucial points are still missing and could be improved prior publication:

Q1- it seems not perfectly adequate to repeat some experiments using the same samples to reach good statistics.

The new “N” numbers were due to the inclusion of more mice for each experiment. None of the previous samples/tissues were used to increase the “N number.

Q2- The role of SLN in Duchenne muscular dystrophy is clearly missing. This could have been addressed by the authors for a more compelling story.

The primary focus of the manuscript is to determine the role of SLN upregulation in DMD. We have previously published a descriptive study demonstrating that the SLN level is upregulated in DMD mouse models. The major goal of the current study is to determine the role of SLN upregulation in DMD by loss-of function approaches. The data obtained from gene knockout mouse models and AAV mediated postnatal suppression of SLN expression clearly demonstrated that SLN upregulation is linked to SERCA inhibition and calcium mishandling and contributes to muscle pathology in DMD. This point is discussed in the manuscript.

Reviewer 3 comments

The revised manuscript NCOMMS-17-03326A by Voit, Babu, et al. (Reducing Sarcolipin Expression Mitigates Duchenne Muscular Dystrophy and Associated Cardiomyopathy in Mice) is a tour-de-force of molecular physiology and translational research. The authors performed additional experiments and revised the manuscript per reviewers’ comments; as such, the revised manuscript is much improved. Additional major comments (2) and minor comments (10) are provided below.

Major comments

Q1. The manuscript neglects a key control (Western blot of myoregulin, MLN) needed to

interpret results of SERCA functional assays and SLN regulation. The mode of regulation detected in Fig 1C (super-inhibition of SERCA K_{Ca} and V_{max}) is most reminiscent of SERCA super-inhibition by SLN and PLB, as induced by formation of a super-inhibitor ternary complex, where calcium affinity is decreased 5-fold and maximal velocity decreased 25-50% in SERCA-PLB-SLN (MacLennan JBC 2002; see also Tupling PLOS 2013). Olson et al. have proposed that MLN is the key regulator in mouse muscle and that SLN plays only a very minor role (Cell 2015, Sci Signal 2016). For example, MLN is robustly expressed in WT mice in quadriceps, plantaris, gastroc, EDL, TA, diaphragm, and soleus (Cell 2015), whereas SLN was undetected in WT mice in quadriceps, plantaris, gastroc, EDL, and TA. Yes, the current NatComm manuscript does demonstrate a large relative upregulation of SLN protein in 3 mouse muscles (quads, pecs, diaphragm), but the current manuscript does not provide absolute quantitation of SLN. (Is SLN upregulated to the same protein level as SERCA? Does the SLN/SERCA stoichiometry exceed 3-5 to allow for full regulation, as demonstrated in heterologous and reconstitution systems?) The functional data in Fig 1C (WT mice) suggest that SERCA shows high activity in mouse diaphragm (i.e., little-to-no MLN inhibition), yet the experiment needs to be performed to eliminate uncertainty in interpretation: Do mouse diaphragm, pectoralis, and quadriceps express MLN, and how do these levels change in mdx, DKO, and TKO mice? Note: FabGennix now sells the first-commercially available MLN antibody, which is reactive to rodent MLN.

The Reviewer's point is well taken and these studies require rigorous biochemical analysis. In the current study, we are unable to demonstrate the absolute stoichiometry of SLN/SERCA and its relation to Ca²⁺ uptake function in various muscles. This is mainly due to the technical limitations such as difficulties in purifying SLN and various SERCA isoforms as standards for quantitation, difference in the sensitivity and specificity of each antibody (SLN, SERCA1, SERCA2a and SERCA2b), and difference in western blot conditions including gel conditions, transfer time and amount of protein loading for each tissues. We therefore provide only the relative expression levels of SLN and SERCA isoforms in each muscle comparison to that of various genotypes. The biochemical analyses that are needed to address these questions require a more detailed and time consuming approach that are beyond the scope and purpose of the current study.

The Reviewer also points out that there is a new MLN antibody available from FabGennix. However, according to the manufacturer, this antibody has not been tested in mouse tissues. We feel that testing this antibody for the specificity of mouse MLN will delay the publication of our work. We would also like to point out that the knowledge of MLN level in dystrophic muscles only provides descriptive information. We agree that the experiments required to elucidate the role MLN in SERCA function in dystrophic muscles is of great interest but require extensive studies similar to the ones performed in the current manuscript including the knockdown of MLN in the presence and absence of SLN.

These points are now discussed appropriately in the manuscript. Please see page #10, 1st paragraph in "Discussion" section.

Q2. Functional effects are typically correlated with expression level of catalytic (SERCA) and regulatory (SLN) subunits. The manuscript does not compare expression levels of SLN or SERCA between mouse tissues, and thus leaves an existing information gap in the literature that the manuscript is poised to finally cover (see Reviewer 3 Comment 5E on the original

manuscript submission).

Data from our previous studies (Schneider et al, J Muscle Res Cell Motil. 2013. Dec;34(5-6):349-56. PMID:23748997) and the current manuscript have shown the relative expression levels of SLN and SERCA proteins in atria, ventricle and some of the skeletal muscle tissues of wildtype and DMD mice and compared to the SR Ca²⁺ uptake function. As stated in our previous response, we are unable to determine the absolute expression levels due to difficulties in purifying SLN and various SERCA isoforms as standards for quantitation, difference in the sensitivity and specificity of each antibody (SLN, SERCA1, SERCA2a and SERCA2b), and difference in western blot conditions including gel conditions, transfer time and amount of protein loading for each tissues. We greatly appreciate the comments of the Reviewer in this regard and we will attempt to compare the absolute ratio of SERCA and its regulators including SLN, MLN in various DMD mouse tissues in our future studies.

Minor comments

Q1. Ca uptake is only measured for one mouse muscle tissue: diaphragm, a 'more affected' tissue. Ca uptake is also measured for mouse ventricle and dog ECU, although EC50 is not reported for mouse ventricle (3 mouse lineages). Addition of a Table summarizing Vmax and EC50 for the different tissues and mouse lineages would be helpful for the reader, as would addition of a 'Ca uptake' entry to the new list "Tissues Used" in the Methods section. Perhaps force frequency fits of CSA-normalized force and half-maximal stimulation frequency, per Reviewer 1 Comment 7, could also be added to the Ca uptake table, as the force parameters are currently only listed in text.

As suggested, the V_{max} and EC₅₀ values are now presented as bar diagrams. Please see new Fig. 1d and 1e, and Supplementary Fig. 1d, 1e, 2d and 2e.

Q2. Ca uptake measurement and analysis: the data are presented as activity vs pCa (Figs 1C, S1C, and S1F), yet the enzyme parameter KCa (Hill fit of Fig 1C graph) is reported as nanomolar (page 4 for S1C, page 5 for Fig 1C). Please pick one unit of Ca concentration (pCa or nM) and use it consistently. Page 4 lines 82-82 incorrectly define EC50 as the 'half maximal velocity of Ca uptake'. The data in Fig 1C are indeed remarkable: ~2.5-fold greater inhibition of SERCA KCa and Vmax in mdx:utr-/- mice (DKO) compared to WT mice, an effect which is removed by knock-out of one SLN allele!

For consistency, we now used "nM" as a unit for Ca²⁺ uptake studies. The pCa in the Ca²⁺ uptake graph is now changed to "nM". EC₅₀ definition is removed.

Q3. SR Ca uptake Methods section on page 17: There appear to be mistakes in the reported amount of ATP and Ruthenium Red added to the assay. 5 mmole of ATP in 1.5 mL = 3.3 M ATP, while 1 umole of RuR in 1.5 mL = 0.67 mM.

We have now provided the final concentration of ATP and Ruthenium Red in the SR Ca²⁺ uptake assays (please refer to "Methods" section).

4. Fig 1d,e,f. Knockout of one SLN gene allele in DKO mice (*mdx:utr*^{-/-}) restores SLN expression to normal in diaphragm (similar to WT muscle = almost undetectable SLN expression) and in pectoralis and quadriceps (similar to WT muscle = no SLN expression). SLN protein expression is dramatically upregulated (>10-fold) in all three muscles of DKO mice. Why is there such a dramatic effect upon knockout of 1 SLN allele (i.e., complete reversal of SLN protein expression) instead of a graded 50% reduction upon knockout of 1 SLN gene in *sln* +/- mice? This is an unexpected result; perhaps it could be corroborated by qRT-PCR, which would quantitate SLN mRNA level (spliced transcripts).

We greatly appreciate this comment and indeed the detailed molecular mechanisms on the regulation of the SLN protein level are not fully understood. This can happen at multiple stages such as transcription efficiency, transcript stability, translation efficiency, post-translational modification and protein half-life. There may also exist complicated positive and negative feedbacks. For example, we have previously made transgenic mice to express a T5A modified SLN under transcriptional regulation of the cardiac specific alpha-MHC promoter. This promoter has been used by many investigators to drive supra-physiological expression of various endogenous genes. Interestingly, in all our transgenic lines, the total SLN level in the atria remained unchanged despite the genomic integration of multiple copies of the T5A SLN expression cassette (Shanmugam et al PLoS One. 2015 Feb 11;10(2):e0115822. doi: 10.1371/journal.pone.0115822. eCollection 2015. PMID: 25671318). As the Reviewer points out, our study is already a “*tour-de-force of molecular physiology and translational research*”. While additional in-depth studies are of interest and can be performed to elaborate the observed dramatic reduction of the SLN protein levels in *mdx:utr*^{-/-}:*sln*^{+/-} mice, these studies are planned for future investigations and their absence in the current investigation does not detract from the primary goal of the study, i.e. reducing SLN level can significantly ameliorate DMD. These points are now discussed appropriately in the manuscript. Please see page #11, 2nd paragraph in “Discussion” section.

Q5. Page 9, lines 237-238 has an unsupported claim and citation. “In contrast, in larger mammals SLN is expressed in all skeletal muscle tissues³³.” The human body contains hundreds of muscles, so it is unlikely that ‘all’ have been tested and confirmed to express SLN. There are many types of ‘larger mammals’, and the SLN global expression pattern in one larger mammal probably does not occur the same for all. The largest animal assayed by SLN Western blot in Ref 33 is rat, with 3 muscles tested: quadriceps, soleus, and diaphragm (Babu et al JMCC 2007), which probably does not serve as an exact model for SLN expression in all human muscles.

Studies shown in Ref# 33 (Babu et al J Mol Cell Cardiol. 2007: 43(2):215-22) compared the expression levels of SLN protein in atria, ventricle, gastrocnemius, soleus, EDL and diaphragm muscles of mouse, rat, rabbit and dog. We have modified the statement to “In contrast, in larger mammals SLN is expressed in all skeletal muscle tissues that have been evaluated”. Please see page #9, 2nd paragraph, 5th line.

Q6. Page 10, line 409: “mS” Does this abbreviation stand for millisecond or milli Siemen?

mS stands for millisecond and it is now spelled-out in the text.

Q7. The manuscript added the number of mice assayed (n) to the middle of each bar in ~half of all the bar graphs in the manuscript (n was added to 17 of 36 bar graphs). Please add 'n' to all bar graphs in manuscript (looks good, useful information, effectively conveyed).

As suggested, we have added *n* number to each bar graph.

Q8. There remain small formatting mistakes throughout the manuscript (missing spaces, extra spaces, missing degree symbol, non-superscripted '3' on cm cubed, etc).

The manuscript was automatically formatted for spacing resulting in missing spaces and extra spaces. Missing degree symbol and superscript mistakes are now corrected.

Q9. Proteolytic fragment of SERCA. Reviewer 3, Comment 9. This is probably a proteolytic fragment of SERCA, and not a non-specific immunoreactive band (as suggested in revision letter), since the band is seen on two blots with two different isoform specific antibodies (SERCA1 and 2). For further proof (in the future for the investigators), additional antibodies should be tested (either pan-SERCA or isoform specific), which is easily performed, since dozens of SERCA antibodies are commercially available.

Thank you for the suggestion. In future experiments, we will utilize other pan-SERCA antibodies to study the SERCA proteolysis.

Q10. The reviewers thank the authors for additional information on their published articles on quantitation of SLN mRNA and protein expression in various species. Quite complex patterns. It's been 9+ years since a SLN review written by Babu was published, and the pace of the SLN field has been accelerating, so it's probably time for another Babu review on SLN-SERCA.

Thank you for your suggestion. We are indeed in the process of writing a review.